# Model Editing for Vision Transformers

**Xinyi Huang**
Independent Researcher
xinyi006@e.ntu.edu.sg

**Kangfei Zhao**
Beijing Institute of Technology
zkf1105@gmail.com

**Long-Kai Huang**[†]
Hong Kong Baptist University
longkai@comp.hkbu.edu.hk

## Abstract

Model editing offers a promising paradigm for efficiently and precisely updating knowledge in pre-trained transformers without costly retraining. While extensively studied in language models (LMs), model editing for vision transformers (ViTs) remains underexplored. Existing methods typically adapt LM-based techniques by modifying the multi-layer perceptron (MLP) modules, overlooking the unique characteristics of ViTs. In this work, we show that ViT predictions are more strongly influenced by the multi-head self-attention (MSA) modules than by the MLPs. Building on this observation, we propose a two-stage framework for editing ViTs. First, we identify which attention heads are most responsible for incorrect predictions. Next, we selectively remove the corresponding features to correct the model's prediction. To further balance error correction with predictive stability on unrelated data, we learn a projection matrix that refines the image representations. Extensive experiments across multiple real-world datasets and model editing benchmarks demonstrate that our method consistently outperforms existing model editing methods for ViTs, achieving superior generalization and locality. Our code is available at https://github.com/shanghxy/Model-editing-for-vision-transformers.

## 1 Introduction

Model editing has emerged as a promising paradigm for efficiently and precisely updating the knowledge encoded in pre-trained transformers, without the need for expensive retraining. While numerous methods have been proposed for editing language models (LMs), there is an equally pressing need to extend these capabilities to vision transformers (ViTs). In real-world deployments, vision models frequently exhibit unexpected prediction failures, as highlighted in recent studies [23, 6, 28, 31]. These failures are particularly common when the downstream data distribution differs from the pre-training distribution. A key reason is that pre-training datasets cannot fully capture the diversity of real-world subpopulations, causing vision models to rely on spurious cues for predictions. For example, models may use background or contextual attributes in images for recognition [34, 20], leading to incorrect predictions when these contexts change. Therefore, there is a strong need for cost-effective model editing approaches to rectify errors in pre-trained vision models.

Editing ViTs, however, presents unique challenges that require specialized solutions. Although LMs and ViTs both use transformer architectures, their training objectives and internal dependencies differ. LMs, trained with auto-regressive loss to predict the next token, primarily encode factual associations in their MLP modules [10, 9, 8, 21]. In contrast, ViTs are trained with a classification loss, and their predictions depend more heavily on multi-head self-attention (MSA) modules, which extract relationships between tokens to form high-level concepts. Our empirical studies (see Sec.3) indicate that MSA modules contribute significantly more to predictions than MLP modules in ViTs, a finding also supported by prior work on CLIP-ViTs [5]. This reveals a key limitation of existing

---

[†]Correpsonding to Long-Kai Huang

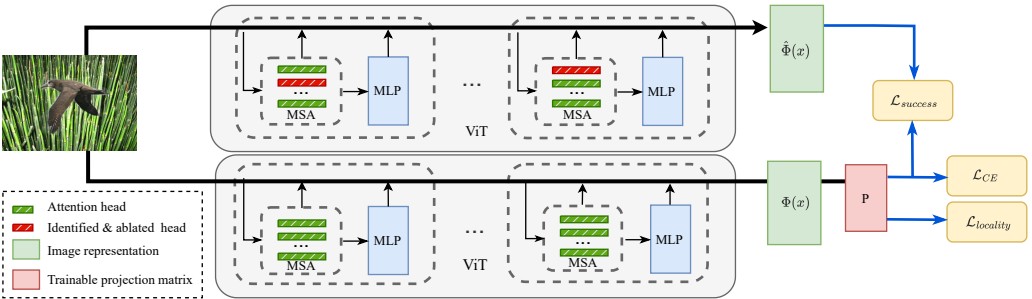

Figure 1: Overview of RefineViT: Stage One identifies attention-head-level features linked to prediction errors. Stage Two learns a projection matrix by optimizing editing success loss ($\mathcal{L}_{success}$), locality loss ($\mathcal{L}_{locality}$), and cross-entropy loss ($\mathcal{L}_{CE}$).

ViT editing methods, such as LWE-ViTs [32], which adapt ideas from LMs by focusing on editing MLP modules. Since ViTs' predictions rely mainly on attention mechanisms rather than factual associations, editing the MLP modules may force the model to overfit the editing data, introducing unexpected information and leading to suboptimal performance. Therefore, specialized solutions that target the unique architectural and functional characteristics of ViTs are necessary for effective model editing.

In this paper, we propose a novel model editing method tailored for ViTs, aimed at rectifying predictive errors using a limited number of samples. Our method is built on the observation that individual attention heads within MSA modules specialize in extracting features related to specific semantic concepts [5]. These modules can detect various concepts from input images and integrate these features into the residual stream. This suggests that classification errors in ViTs often arise not from a failure to capture essential features, but from an over-reliance on spurious or non-causal ones.

To correct the prediction errors, we decompose the final representation of ViTs into features at both the MLP-layer level and attention-head level. Leveraging this decomposition, we develop a two-stage framework called RefineViT. In the first stage, we identify attention-head-level features that contribute to prediction errors by analyzing a small set of samples. In the second stage, we rectify the final representation by selectively ablating these problematic features. The framework enables effective model editing with only a few examples and is illustrated in Fig. 1.

The second stage, representation rectification, can be implemented in different ways depending on the task. For binary classification, we find that simply zeroing out the identified attention-head features yields substantial performance gains. However, in multi-label classification, directly removing these features may degrade performance on unrelated classes, as the ablated heads may still be important for other predictions. To address this issue, we learn a projection matrix that aligns the final representation of erroneous samples with their ablated counterparts, while preserving the original representations of correctly predicted data. This approach balances error correction with predictive stability, allowing precise model edits while minimizing side effects. The projection can be integrated into the classifier, requiring no changes to the model architecture. Notably, our framework can also be seamlessly applied to ViT-based models like CLIP-ViT, which use cosine similarity between image and text embeddings for classification.

**Summary of contribution:**

- We empirically reveal a key difference between LMs and ViTs for model editing: while editing MLP layers is effective in LMs, ViT predictions depend mainly on MSA modules, and errors are often driven by spurious features extracted by specific attention heads. This suggests that effective ViT editing should target MSA modules rather than MLPs.

- We propose a two-stage method that identifies and removes problematic attention-head features from the final representation. To ensure both editing success and locality in multi-label classification, we learn a projection matrix that aligns erroneous samples with their corrected counterparts while preserving correct predictions.

- We validate the effectiveness of our framework on multiple real-world datasets and model editing benchmarks. Our method achieves strong error correction with minimal impact on unrelated data,

outperforming existing ViT editing baselines in both generalization and locality. Moreover, it generalizes well to ViT-based architectures such as CLIP-ViT.

## 2 Preliminaries

### 2.1 Problem Formulation: Model Editing for ViTs

In this paper, we tackle the challenge of error correction in Vision Transformers (ViTs). When a ViT misclassifies an image $\boldsymbol{x}$ as $\hat{y}$ instead of its true label $y$, this error often reflects a broader pattern rather than an isolated mistake. Such errors typically recur across similar images with the same true label $y$. We define this subset of misclassified data as $\mathcal{I}_{y,\hat{y}} = \{\boldsymbol{x}'|f_\theta(\boldsymbol{x}') = \hat{y}\}$, where $f_\theta$ is the ViT model with parameters $\theta$.

Model editing aims to fix these misclassifications by refining the model to correct not only the specific image $\boldsymbol{x}$ but also similar instances in $\mathcal{I}_{y,\hat{y}}$. This process focuses on two key goals:

- *Edit success* measures how well the modified model correctly classifies previously misclassified data. We evaluate this by measuring the edited model's accuracy on $\mathcal{I}_{y,\hat{y}}$.
- *Edit locality* measures how well the modified model preserves the prediction on unrelated data. This is measured by the edited model's accuracy on data outside the target subset: $\mathcal{O}_{y,\hat{y}} = \{\boldsymbol{x}|\boldsymbol{x} \notin \mathcal{I}_{y,\hat{y}}\}$. High edit locality indicates that the editing has minimal impact on unrelated data.

In practice, failures in object recognition are typically identified by testing the model's performance on a small set of samples from each class. These samples, along with their true labels and model predictions, serve as the foundational data for model editing.

### 2.2 ViT Image Representation Decomposition

A Vision Transformer (ViT) consists of $L$ residual blocks. Each block contains a multi-head self-attention (MSA) layer, followed by a multi-layer perceptron (MLP) layer. Both layers are preceded by layer normalization. The ViT processes an input image $\boldsymbol{x}$ by dividing it into $N$ patches. Each patch is embedded into a $d$-dimensional token, resulting in $N$ tokens: $\{\boldsymbol{z}_i^0\}_{i=1}^N$. Additionally, a special class token $\boldsymbol{z}_{cls}^0$ is added. Together, these form the initial state of the residual stream $\boldsymbol{Z}^0 = [\boldsymbol{z}_{cls}^0, \boldsymbol{z}_1^0, ..., \boldsymbol{z}_N^0] \in \mathbb{R}^{d\times(N+1)}$, which is updated by the residual blocks. The residual blocks sequentially update this stream, and the layer-normalized class token from the final layer serves as the image representation $\Phi(\boldsymbol{x})$:

$$\hat{\boldsymbol{Z}}^l = \text{MSA}^l\left(\boldsymbol{Z}^{l-1}\right) + \boldsymbol{Z}^{l-1}, \ \ \boldsymbol{Z}^l = \text{MLP}^l\left(\hat{\boldsymbol{Z}}^l\right) + \hat{\boldsymbol{Z}}^l, \ \ l \in \{1, 2, ..., L\}. \tag{1}$$

$$\Phi(\boldsymbol{x}) = [\boldsymbol{Z}^L]_{cls} = [\boldsymbol{Z}^0]_{cls} + \sum_{l=1}^L [\text{MSA}^l(\boldsymbol{Z}^{l-1})]_{cls} + \sum_{l=1}^L [\text{MLP}^l(\hat{\boldsymbol{Z}}^l)]_{cls}, \tag{2}$$

where $[\boldsymbol{Z}^L]_{cls}$ refers to the first column of $\boldsymbol{Z}^L$, corresponding to the class token. Here, we omit the layer normalization for notational simplicity.

Following [4], the output of each MSA layer can be written as the sum of the outputs from all attention heads, each multiplied by its own output matrix $\boldsymbol{W}_O^h$. Specifically, the class token output from the $l$-th MSA layer can be decomposed as:

$$[\text{MSA}^l(\boldsymbol{Z}^{l-1})]_{cls} = \sum_{h=1}^H \left[\text{Head}^{l,h}\left(\boldsymbol{Z}^{l-1}\right)\right]_{cls} = \sum_{h=1}^H \sum_{i=0}^N a_{0,i}^{l,h} \boldsymbol{W}_O^{l,h} \boldsymbol{W}_V^{l,h} \boldsymbol{z}_i^{l-1} \tag{3}$$

Here, $H$ is the number of attention heads in each layer; $\text{Head}^{l,h}$ is the $h$-th head in the $l$-th MSA layer; $a_{0,i}^{l,h}$ is the attention weights from the class token to the $i$-th token; $\boldsymbol{W}_O^{l,h}$ and $\boldsymbol{W}_V^{l,h}$ are the output and value projection matrix; $\boldsymbol{z}_i^{l-1}$ is the $i$-th token from the $(l-1)$-th MLP layer.

By substituting Eq. (3) into Eqs. (1)-(2) and defining $\boldsymbol{h}^{l,h} = [\text{Head}^{l,h}\left(\boldsymbol{Z}^{l-1}\right)]_{cls}$ for simplicity, we can express the final image representation as a sum of contributions from the initial class token, all

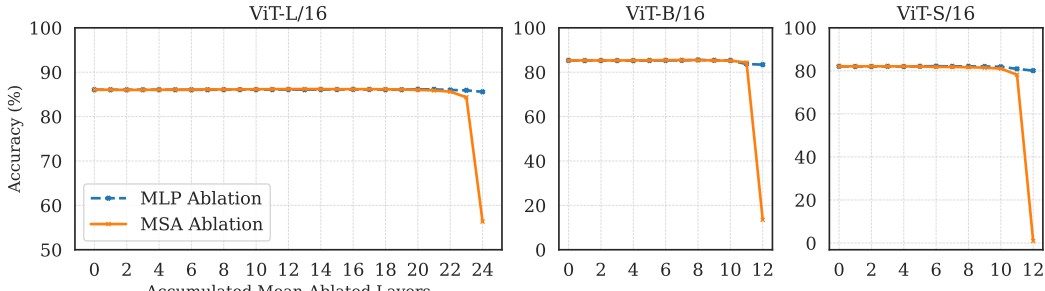

Figure 2: Accuracy changes after mean ablation of MLP and MSA modules.

MLP layers, and all attention heads:

$$\Phi(\boldsymbol{x}) = [\boldsymbol{Z}^0]_{cls} + \sum_{l=1}^{L}[\text{MLP}^l(\hat{\boldsymbol{Z}}_l)]_{cls} + \sum_{l=1}^{L}\sum_{h=1}^{H}\boldsymbol{h}^{l,h}. \tag{4}$$

The decomposition in Eq. (4) illustrates the direct contribution of the initial class token, each MLP layer, and each attention head in the MSA layers to the final image representation [5, 1].

## 3 Differences in Model Editing between LMs and ViTs

Previous studies [22, 21, 10, 9] suggest that feed-forward layers in Transformers play a crucial role in shaping predictions, making them effective targets for model editing. However, these findings are largely limited to Language Models (LMs) and may not be generalized to Visual Transformers (ViTs). In this section, we analyze the roles of MSA and MLP modules in ViTs, and highlight key differences in model editing between LMs and ViTs.

### 3.1 Direct Effects of MSA and MLP Modules in ViTs

To quantify the direct effect of each component, we measure the drop in classification accuracy after mean ablation. Specifically, we replace the output features of target modules with their dataset-wide mean, recompute the representations (see Eq. 4), and evaluate the resulting accuracy. A larger drop indicates a stronger direct effect on predictions.

We conduct this analysis on ViT-S/16, ViT-B/16, and ViT-L/16 using 10,000 randomly sampled images from the ImageNet test set [3]. These samples are used to compute mean values for each component and to assess classification accuracy.

Fig. 2 presents classification accuracies when replacing all direct contributions of MSAs or MLPs up to a given layer with their dataset-wide mean. Two key observations emerge. First, ablating MLPs causes only a marginal drop in accuracy, indicating that MLPs have a minimal direct impact on predictions. Second, the effect of ablating MSAs varies across layers: replacing early MSAs has little effect, while modifying the last two leads to a notable accuracy decline, highlighting their critical role in final predictions.

These findings suggest that, unlike in LMs, editing MLP layers in ViTs is less likely to yield significant changes in predictions. Therefore, alternative strategies should be explored for effective model editing in ViTs.

### 3.2 The Devil is in the Spurious Correlations

Based on these observations, we hypothesize that many errors in ViTs are caused by the model relying on misleading features, a phenomenon known as spurious correlation [7, 24, 11]. Simply fine-tuning MSA modules using the edit sample can distort the features extracted by attention heads, affecting predictions on unrelated data and resulting in low edit locality. A more effective strategy may be to remove the influence of misleading features only for the target samples.

To test this hypothesis, we conduct an empirical study to examine whether removing the direct contribution of a feature from the image representation can facilitate error correction. We use the

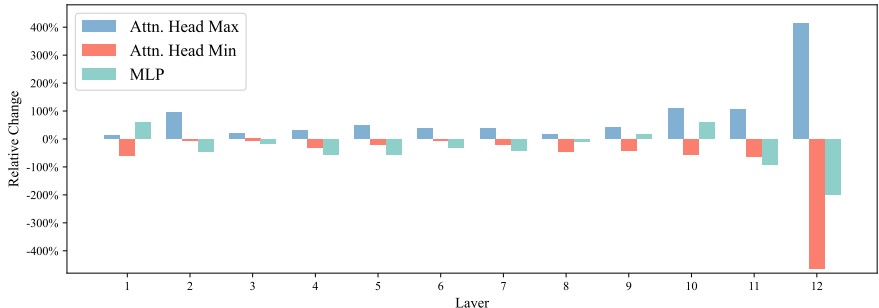

Figure 3: Relative change in the logit gap between ground-truth and incorrect classes after ablating attention heads or MLP module at different layers. For attention heads, we show the maximal and minimal relative changes after ablating four heads. Results are based on ViT-B/16.

same editing dataset as in [32]. For each editing case, we remove the direct effect of the feature extracted by a specific component, such as an MLP module or an attention head in MSA modules, and report the relative change in the logit gap between the ground-truth class and the incorrectly predicted class. Specifically, let $g_o$ and $g$ denote the logit gaps between the ground-truth class and the incorrectly predicted class before and after feature removal, respectively. The relative change is calculated as $g - g_o/|g_o|$. A relative change greater than $100\%$ indicates that the prediction is likely to shift from the original incorrect class to the ground-truth class.

The results in Fig. 3 show that ablating certain attention heads can achieve a positive relative change, even greater than $100\%$. This supports our hypothesis that errors are often caused by misleading features extracted by specific attention heads, and that removing their direct effect can correct these errors. Additionally, we observe that ablating attention heads has a much greater impact than ablating MLP modules, further confirming the central role of MSA modules in ViT predictions.

## 4 RefineViT: Model Editing for ViT

In the previous section, we show that misclassifications in ViTs often stem from misleading features extracted by attention heads. Building on this observation, we propose RefineViT—a two-stage model editing framework designed to mitigate such biases. In the first stage, we identify the attention heads most responsible for misclassification. In the second stage, we temporarily disable (ablate) them to create a modified model that corrects the error. To improve edit locality, we use the knowledge gained from this ablation to update the original model, enabling it to fix the error while preserving predictions on unrelated inputs.

### 4.1 Identify the Cause of Errors

Our analysis shows that attention heads in the later MSA layers have a strong direct impact on the model's final predictions. Accordingly, we focus on these layers to find the specific heads most responsible for misclassification.

To this end, we define an ablation matrix $\boldsymbol{A}$, where each element $A_{lh} \in \{0, 1\}$ indicates whether the $h$-th head in $l$-th layer is ablated ($A_{lh} = 1$) or not ($A_{lh} = 0$). When a head is ablated, its feature is replaced by zero in the representation. Therefore, the image representation after ablation $\hat{\Phi}(\boldsymbol{x}, \boldsymbol{A})$ is calculated as

$$\hat{\Phi}(\boldsymbol{x}, \boldsymbol{A}) = [\boldsymbol{Z}^0]_{cls} + \sum_{l=1}^{L}[\text{MLP}^l(\hat{\boldsymbol{Z}}^l)]_{cls} + \sum_{l=1}^{L}\sum_{h=1}^{H}(1 - A_{lh})\boldsymbol{h}^{l,h} = \Phi(\boldsymbol{x}) - \sum_{l=1}^{L}\sum_{h=1}^{H}A_{lh}\boldsymbol{h}^{l,h}, \quad (5)$$

where $\Phi(\boldsymbol{x})$ is the original image representation.

To measure how effective an ablation is at correcting an error, we define the following utility function:

$$U(\boldsymbol{x}, y, \hat{y}, \boldsymbol{A}) = f\left(\hat{\Phi}(\boldsymbol{x}, \boldsymbol{A})\right)_y - f\left(\hat{\Phi}(\boldsymbol{x}, \boldsymbol{A})\right)_{\hat{y}}, \quad (6)$$

where $\boldsymbol{x}$ is the input image, $y$ is the ground-truth label, $\hat{y}$ is the incorrect prediction, and $f(\cdot)_y$ represents the output logit for class $y$ from the pretrained classifier.

From the perspective of ablation study, $U(\boldsymbol{x}, y, \hat{y}, \boldsymbol{A}) - U(\boldsymbol{x}, y, \hat{y}, \boldsymbol{0})$ simultaneously quantifies the of direct effect of the ablated attention heads to the incorrectly predicted label $\hat{y}$ as $f\left(\Phi(\boldsymbol{x})\right)_{\hat{y}} - f\left(\hat{\Phi}(\boldsymbol{x}, \boldsymbol{A})\right)_{\hat{y}}$ and the negative direct effect to the ground-truth label $y$ as $f\left(\Phi(\boldsymbol{x})\right)_y - f\left(\hat{\Phi}(\boldsymbol{x}, \boldsymbol{A})\right)_y$. Since $U(\boldsymbol{x}, y, \hat{y}, \boldsymbol{0})$ is constant, we can ignore it and identify the attention heads within these layers that contribute most to prediction errors by maximizing this utility

$$\boldsymbol{A}^* = \arg\max_{\boldsymbol{A}} \quad U(\boldsymbol{x}, y, \hat{y}, \boldsymbol{A}). \tag{7}$$

However, solving the problem regarding the binary matrix $\boldsymbol{A}$ is NP-hard and generally requires searching over all the possible solutions, which is infeasible. To make this tractable, we consider ablations where only one head is ablated at a time. For each head, we compute the utility $U(\boldsymbol{x}, y, \hat{y}, \mathbb{1}_{lh})$, where $\mathbb{1}_{lh}$ is a matrix with a single 1 at position $(l, h)$ and 0 elsewhere. We then rank all $L \times H$ heads by their utility scores and select the top $T_h$ heads with the highest scores. Empirically, we observe that the selected heads are predominantly located in the later layers, consistent with our earlier findings. We denote the set of these heads as $\mathcal{S}_{\text{ablate}}$ and construct the approximate optimal ablation matrix $\hat{\boldsymbol{A}}$ by setting $\hat{A}_{lh} = 1$ for $(l, h) \in \mathcal{S}_{\text{ablate}}$ and 0 otherwise.

In most cases, only a single error sample is available for editing, so we use zero-ablation to estimate the contribution of each attention head. When more error samples or correctly classified samples are available, more advanced techniques, such as mean ablation, can be used for a finer analysis. Details of these extended methods are provided in Appendix A.

## 4.2 Refine ViT Through Representation Rectification

After identifying the attention heads that cause errors, a straightforward way to fix mistakes is to directly remove their direct contributions from the image representation. For binary classification tasks, this simple ablation can already lead to large improvements. Specifically, we compute the ablated image representation as $\hat{\Phi}(\boldsymbol{x}, \hat{\boldsymbol{A}})$.

However, in more complex settings such as multi-label classification, simply removing these features can hurt performance on other classes. This is because the ablated heads may still be important for unrelated predictions. Moreover, modifying the forward pass to remove features can make deployment more difficult.

To address these issues, we introduce a feature projection matrix $\boldsymbol{P}(\theta) \in \mathbb{R}^{d \times d}$, where $\theta$ is the trainable parameters and $d$ is the dimension of the image representation. This matrix is applied after the Transformer blocks and rectifies the image representation as follows:

$$\Phi^{\text{proj}}(\boldsymbol{x}) = \boldsymbol{P}(\theta)\Phi(\boldsymbol{x}) = \boldsymbol{P}(\theta)[\boldsymbol{Z}^0]_{cls} + \sum_{l=1}^{L} \boldsymbol{P}(\theta)[\text{MLP}^l(\hat{\boldsymbol{Z}}^l)]_{cls} + \sum_{l=1}^{L}\sum_{h=1}^{H} \boldsymbol{P}(\theta)\boldsymbol{h}^{l,h}. \tag{8}$$

To achieve edit success, we propose transferring knowledge from the ablated representation, $\hat{\Phi}(\boldsymbol{x}, \hat{\boldsymbol{A}})$, to the projected representation by minimizing the MSE loss for samples where the ViT fails to make accurate predictions ($\boldsymbol{x} \in \mathcal{D}_{\text{failed}}$) as follows:

$$\mathcal{L}_{\text{success}}(\theta) = \mathbb{E}_{\boldsymbol{x} \in \mathcal{D}_{\text{failed}}} \left\| \boldsymbol{P}(\theta)\Phi(\boldsymbol{x}) - \hat{\Phi}(\boldsymbol{x}, \hat{\boldsymbol{A}}) \right\|^2. \tag{9}$$

To achieve edit locality, we want the projection to keep the representation unchanged, preserving the model's original behavior. We use another MSE loss to align the representations before and after projection for samples where the model is already correct ($\boldsymbol{x} \in \mathcal{D}_{\text{success}}$) as follows:

$$\mathcal{L}_{\text{locality}}(\theta) = \mathbb{E}_{\boldsymbol{x} \in \mathcal{D}_{\text{success}}} \left\| \boldsymbol{P}(\theta)\Phi(\boldsymbol{x}) - \Phi(\boldsymbol{x}) \right\|^2. \tag{10}$$

If $\mathcal{D}_{\text{success}}$ is not available, we constrain the projection matrix to be close to the identity matrix $I$ as

$$\mathcal{L}_{\text{locality}}(\theta) = \left\| \boldsymbol{P}(\theta) - I \right\|_F^2. \tag{11}$$

We combine these two objectives with the standard cross-entropy loss $\mathcal{L}_{\text{CE}}(\theta)$ for classification. The total loss for learning the projection is:

$$\theta^* = \arg\min_{\theta} \left[ \alpha\mathcal{L}_{\text{success}}(\theta) + \beta\mathcal{L}_{\text{locality}}(\theta) + \mathcal{L}_{\text{CE}}(\theta) \right], \tag{12}$$

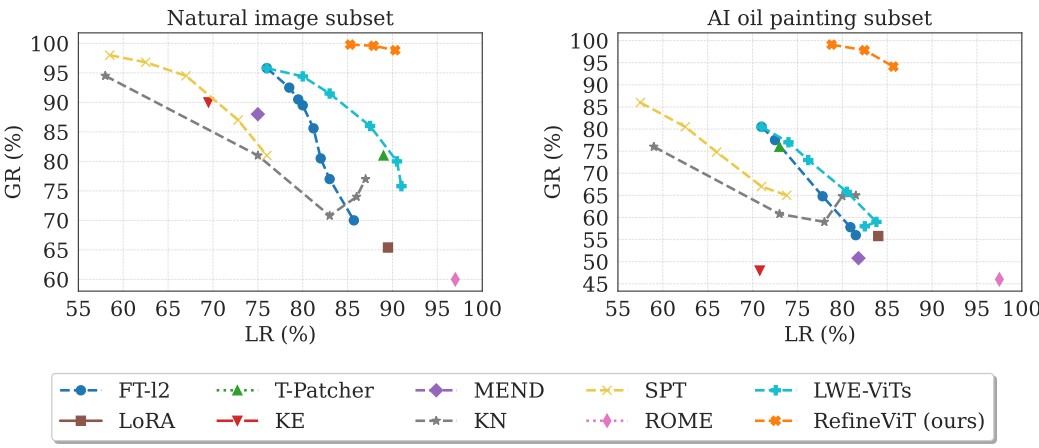

Figure 4: Editing results for ViT/B-16.

where $\alpha$ and $\beta$ are hyperparameters that balance error correction and locality preservation.

This projection-based approach allows us to correct specific errors while minimizing side effects on other predictions. Importantly, it can be easily integrated into ViT-based models, including those like CLIP-ViT that use cosine similarity for classification, since the projection only modifies the image representation before the classifier.

# 5 Related Work

Model editing techniques [22, 2, 33] aim to refine the behavior of LLMs for specific input-output pairs, while preserving their performance on other data. These methods can be grouped into three main categories: classifier-based, meta-learning-based, and locate-then-edit methods.

Classifier-based model editing methods retain the pre-trained parameters and use an auxiliary classifier to determine behavioral modifications. This method ensures that the modifications are applied only to targeted samples, leaving the original model predictions unchanged for unrelated samples outside the edited scope. Locate-then-edit methods first identify model parameters associated with specific knowledge, often through causal tracing, and then directly update these parameters to achieve the desired edits. Meta-learning-based methods utilize a hyper-network, known as an editor, to update parameters. This editor is meta-trained across multiple editing tasks to learn how to generate the necessary updates based on the provided edit samples. For a comprehensive review of these methods for language models, please refer to [33].

Despite its strides in language models, adapting similar techniques to visual models like Vision Transformers (ViTs) and CLIP remains largely untapped. [26] adapted classifiers in convolutional neural networks to mitigate concept-level spurious features by mapping misleading visual concepts to correct targets. However, this requires prior knowledge of the erroneous visual concept, its location, and the target concept, which may not always be available. Another line of work [1, 5] proposes to interpret model components and ablate spurious components to rectify errors. Yet, these methods also rely on knowing which visual concept triggers the error. Recently, [32] introduced a method inspired by LMs editing techniques, which fine-tunes MLP modules in ViTs and employs a hyper-network to determine where to apply edits. However, as discussed in Section 3, this approach may yield suboptimal performance due to fundamental differences between LMs and ViTs.

Our method shares the goal of correcting errors by removing misleading features, as in [1, 5, 16, 26]. However, unlike these approaches, we identify spurious features using a data-driven strategy, without requiring prior knowledge of biases or their locations. This makes our method more practical for real-world scenarios where such information is not readily available.

Table 1: Average-group accuracy (%) and worst-group accuracy (%) on the Waterbirds Dataset. Methods marked with an asterisk (*) use additional data for training or validation. The best results are highlighted in **bold**, while the second-best results are underlined.

| Method | ViT-B/16 | | ViT-L/14 | | ViT-H/14 | |
|---|---|---|---|---|---|---|
| | Avg. ($\uparrow$) | Wst. ($\uparrow$) | Avg. ($\uparrow$) | Wst. ($\uparrow$) | Avg. ($\uparrow$) | Wst. ($\uparrow$) |
| Base | 72.8 | 45.6 | 75.5 | 47.7 | 68.6 | 37.2 |
| Tip-Adapter (training-free) | 74.4 | 46.9 | 77.4 | 52.6 | 70.3 | 38.0 |
| Tip-Adapter (training-based) | 76.3 | 49.9 | 78.0 | 52.2 | 74.8 | **59.3** |
| RefineViT (ours) | **81.1** | 61.4 | 85.5 | 72.1 | **75.9** | 51.3 |
| TextSpan * | 78.5 | 57.5 | 84.4 | 72.9 | 72.9 | 43.3 |
| RefineViT (ours) * | 80.4 | **65.9** | **85.6** | **75.6** | **75.9** | 51.3 |

# 6 Experiments

We evaluate the proposed method, RefineViT, on the ViT editing benchmark from [32], the Binary Waterbirds dataset [25], CelebA [19], ImageNet-R [13], and ImageNet-A [14]. Our experiments are designed to answer the following research questions: **Q1**: Does RefineViT, which edits the MSA modules, outperform state-of-the-art methods that primarily focus on editing the MLP modules, in terms of both editing success and locality in ViTs? (Sections 6.1)   **Q2**: Can RefineViT generalize to other ViT-based models such as CLIP-ViT? (Section 6.2)    **Q3**: Why does RefineViT work? Specifically, is it effective in identifying the attention-head-level features responsible for prediction failures? (Appendix C)   Ablation studies and sensitivity analyses are provided in Appendix D.

## 6.1 Edit success and edit locality

To address **Q1** and assess RefineViT's editing success and locality, we evaluate it on the benchmark proposed by [32]. This benchmark collects misclassified samples by ViTs from naturally underrepresented images and AI-generated images. Further details about this benchmark are provided in Appendix B.1.

**Evaluation Metrics.** Following the experimental setup in [32], we evaluate all model editing methods on the single-sample editing task and compare their performance using three evaluation metrics: 1) *Success Rate (SR)*: the prediction success rate of the edited model on the single sample used for correction; 2) *Generalization Rate (GR)*: the accuracy of the edited model on neighboring samples within the editing scope; 3) *Locality Rate (LR)*: the accuracy of the edited model on unrelated samples outside the editing scope.

**Competing Methods.** We rigorously evaluate RefineViT under the same conditions as [32], and compare its performance with recent model editing methods as follows:**1)** *Learning-Where-To-Edit (LWE-ViTs)* [32]: A meta-learning-based approach that selects edit locations within the 8th to 10th MLP layers in ViT-B/16; **2)** *Standard Fine-Tuning (FT)*: standard fine-tuning targeting the 8th to 10th MLP layers; **3)** *FT-$\ell_2$*: extends standard fine-tuning by incorporating $\ell_2$-norm regularization; **4)** *KE* [2] and **5)** *MEND* [22]: leverage hyper-networks to guide parameter updates in the last three MLP layers; **6)** *T-Patcher* [17]: introduces and trains a small set of additional neurons in the final MLP layer; **7)** *SPT* [12]: sensitivity-aware, parameter-efficient fine-tuning; **8)** *ROME* [21]: modifies the last MLP layer to update specific factual associations; **9)** *LoRA* [15]: low-rank updates to all MSA layers in the Transformer.

**Main Results.** Since most methods achieve nearly 100% *SR*, demonstrating their effectiveness in correcting single predictive errors, we focus on the GR-LR curve in Fig.4. As shown, RefineViT outperforms LWE-ViTs [32], which is the previous state-of-the-art on this benchmark, and achieves substantially better GR-LR performance than all baselines. This improvement likely stems from the fact that most existing methods (*LWE-ViTs, FT, FT-$\ell_2$, KE, MEND, T-Patcher, ROME*) focus on editing MLP layers, a strategy inspired by findings in language models that highlight the efficacy of modifying MLP modules for NLP tasks. In contrast, RefineViT targets the last few MSA layers, which proves more effective for computer vision tasks. Although *LoRA* also targets MSA layers, it updates all of them directly and simultaneously, resulting in a large number of trainable parameters and potential distortion of attention head features.

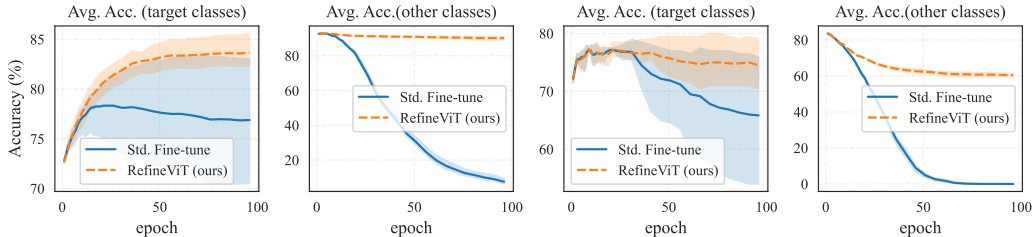

Figure 5: Comparison of RefineViT and Standard Fine-Tuning with ViT-L/14. From left to right, the first two figures show the average accuracy for the target classes and other classes on the Waterbirds + ImageNet-R combined dataset, while the next two figures display the corresponding results for the ImageNet-A dataset. We repeat the experiment with three random seeds and report the average results.

Moreover, RefineViT is highly efficient, as it avoids modifying the Transformer and instead trains a lightweight projection matrix. This allows updates in under 0.3 seconds for 50 epochs on a single NVIDIA A100 (40GB) as the output of the backbone can be cached and reused. In comparison, *LWE-ViTs* takes around 12 seconds for fine-tuning under the same conditions, without considering the extra cost of training a hyper-network before editing.

## 6.2 Generalization to ViT-Based Models

Although RefineViT is designed for ViTs, it generalizes effectively to ViT-based models such as CLIP-ViT, which uses cosine similarity between textual and visual embeddings for zero-shot classification. We evaluate RefineViT within CLIP-ViT for both performance enhancement and model editing tasks.

**Performance Enhancement and Debiasing.** We evaluate RefineViT on the Binary Waterbirds [25] and CelebA [19] datasets, both widely used benchmarks for CLIP debiasing. Our goal is to enhance CLIP-ViTs' zero-shot ability to distinguish between waterbirds and landbirds despite background interference in the Binary Waterbirds dataset, and to improve its accuracy in predicting age (young vs. old) in the CelebA dataset. More implementation details can be found in Appendix B.2

**Competing Methods.** On the Waterbirds dataset, we compare RefineViT with: (1) *TextSpan* [5], which treats background as a known spurious feature and uses human expertise to analyze over 5,000 test images and identify spurious attention heads; and (2) *Tip-Adapter* [35], a robust method that improves CLIP-ViT's accuracy while preserving its zero-shot capabilities. We evaluate both its training-free and training-based variants. For the CelebA dataset, where spurious features are unknown, we compare only against Tip-Adapter. Since these baselines focus on binary classification rather than model editing, we use a simplified version of RefineViT, applying zero ablation in the second stage instead of learning a projection matrix.

**Results.** Table 1 presents the performance of various methods and CLIP-ViT variants on the Binary Waterbirds dataset. We report both average and worst-case accuracy across the four bird groups (landbirds on land, landbirds on water, waterbirds on water, and waterbirds on land). For CelebA, Table 2 shows the results of RefineViT and Tip-Adapter in predicting the 'Young' vs. 'Old' attribute using CLIP-ViT-B/16. Key observations include: (i) Given the same number of samples per class, RefineViT outperforms Tip-Adapter on both datasets. (ii) RefineViT also generally surpasses TextSpan on Waterbirds, despite using fewer samples, incurring lower computational cost, and requiring no human intervention. Its performance can be further improved with a validation set.

**Model Editing.** We evaluate RefineViT for CLIP-ViT in model editing scenarios using ImageNet-A and a combined dataset of Waterbirds and ImageNet-R. Additional details are available in Appendix B.3.

**Results.** We compare RefineViT with standard fine-tuning, which also updates the projection matrix using the same samples. As shown in Fig. 5, RefineViT achieves higher accuracy on target classes while substantially mitigating performance drops on non-target classes, avoiding the common issue caused by overfitting in standard fine-tuning.

**Ablation study and Sensitive Analysis.** We conduct experiments to analyze the impact of the hyperparameters $\alpha$ and $\beta$ in the loss function Eq. (12), the number of candidate attention heads

selected ($T$) in stage one, the update strategies employed, and other factors to fully assess our method. The results and analyses are presented in Appendix D.

## 7 Conclusion and Future Directions

In this work, we study model editing for vision transformers and uncover a key difference from language models. While MLP modules are often the main editing targets in language models, we find that the predictions of vision transformers are more sensitive to changes in the multi-head self-attention layers. Based on this finding, we introduce RefineViT, a two-stage framework that first identifies the attention heads responsible for prediction errors and then refines their representations to correct these errors. Experiments on standard benchmarks show that RefineViT achieves SOTA performance in both edit accuracy and generalization. The framework also generalizes well to ViT-based models such as CLIP-ViT, demonstrating its broader applicability. Additional ablation and sensitivity studies confirm the robustness and effectiveness of the proposed approach.

Our framework still faces several limitations. The current analysis considers only the direct effect (first-order effect) of individual attention heads and does not account for the indirect effect that may accumulate across layers. Incorporating these higher-order effects may further improve reliability. In addition, our experiments focus on image classification tasks. Extending RefineViT to dense-prediction tasks will require new attribution strategies and evaluation protocols. Finally, model editing involves a trade-off between success, generalization, and locality. Although our projection-based refinement helps reduce unintended side effects, achieving complete locality remains an open challenge for future work.

## Acknowledgments and Disclosure of Funding

Kangfei Zhao is supported by National Key Research and Development Plan, No. 2023YFF0725101.

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

# A  Head Identification Using Mean Ablation

The utility function in Eq. (18), which is based on zero-ablation, is designed for the typical model editing scenario where only a single error sample is available. To extend this approach to cases where multiple error or correctly classified samples are present, we introduce a mean-ablation strategy. Specifically, we assume access to three sets of data: (1) a set of misclassified samples with ground-truth label $y$ and predicted label $\hat{y}$, denoted as $\mathcal{W}_{y,\hat{y}}$; (2) a set of correctly classified samples with ground-truth label $y$, denoted as $\mathcal{C}_y$; and (3) a set of correctly classified samples with ground-truth label $\hat{y}$, denoted as $\mathcal{C}_{\hat{y}}$. For each set, we compute the average feature for each attention head, denoted as $\bar{\boldsymbol{h}}_{\mathcal{W}_{y,\hat{y}}}^{l,h}$, $\bar{\boldsymbol{h}}_{\mathcal{C}_y}^{l,h}$, and $\bar{\boldsymbol{h}}_{\mathcal{C}_{\hat{y}}}^{l,h}$, respectively. The mean-ablated representation of an input $\boldsymbol{x}$ is then defined as

$$\tilde{\Phi}(\boldsymbol{x}, \bar{\boldsymbol{h}}^{l,h}) = \Phi(\boldsymbol{x}) - \boldsymbol{h}^{l,h} + \bar{\boldsymbol{h}}^{l,h}. \tag{13}$$

We propose four utility functions to assess the contribution of each attention head to model predictions, each leveraging different combinations of the available data sets.

**Utility A.** To evaluate the influence of attention head $\boldsymbol{h}^{l,h}$, we first consider replacing its feature in misclassified samples $\mathcal{W}_{y,\hat{y}}$ with the average feature from correctly classified samples $\mathcal{C}_y$, i.e., $\bar{\boldsymbol{h}}_{\mathcal{C}_y}^{l,h}$. We then measure whether this replacement shifts the model's prediction towards the correct label $y$ and away from the incorrect label $\hat{y}$:

$$U_A^{l,h} = \mathbb{E}_{\boldsymbol{x} \in \mathcal{W}_{y,\hat{y}}} \left[ f\left(\tilde{\Phi}(\boldsymbol{x}, \bar{\boldsymbol{h}}_{\mathcal{C}_y}^{l,h})\right)_y - f\left(\tilde{\Phi}(\boldsymbol{x}, \bar{\boldsymbol{h}}_{\mathcal{C}_y}^{l,h})\right)_{\hat{y}} \right]. \tag{14}$$

A large value of $U_A^{l,h}$ indicates that the head in question plays a significant role in the model's misclassification, as its replacement leads to a notable correction in the prediction.

**Utility B.** While Utility A focuses on misclassified samples, Utility B takes the opposite perspective by considering correctly classified samples in $\mathcal{C}_y$. Here, we replace the attention head feature with the average from misclassified samples $\bar{\boldsymbol{h}}_{\mathcal{W}_{y,\hat{y}}}^{l,h}$ and measure the extent to which this substitution causes the prediction to deteriorate, i.e., to shift towards the incorrect label $\hat{y}$:

$$U_B^{l,h} = \mathbb{E}_{\boldsymbol{x} \in \mathcal{C}_y} \left[ f\left(\tilde{\Phi}(\boldsymbol{x}, \bar{\boldsymbol{h}}_{\mathcal{W}_{y,\hat{y}}}^{l,h})\right)_{\hat{y}} - f\left(\tilde{\Phi}(\boldsymbol{x}, \bar{\boldsymbol{h}}_{\mathcal{W}_{y,\hat{y}}}^{l,h})\right)_y \right]. \tag{15}$$

A higher value of $U_B^{l,h}$ suggests that the head, when replaced with features from misclassified samples, pushes the model towards making the same error, highlighting its contribution to in incorrect predictions.

**Utility C.** Utilities A and B focus on samples with the same ground-truth label but different predictions. In contrast, Utilities C and D examine samples that share the same predicted label but have different ground-truth labels. For Utility C, we consider misclassified samples in $\mathcal{W}_{y,\hat{y}}$ and replace the attention head feature with the average from correctly classified samples of class $\hat{y}$, i.e., $\bar{\boldsymbol{h}}_{\mathcal{C}_{\hat{y}}}^{l,h}$:

$$U_C^{l,h} = \mathbb{E}_{\boldsymbol{x} \in \mathcal{W}_{y,\hat{y}}} \left[ f\left(\tilde{\Phi}(\boldsymbol{x}, \bar{\boldsymbol{h}}_{\mathcal{C}_{\hat{y}}}^{l,h})\right)_y - f\left(\tilde{\Phi}(\boldsymbol{x}, \bar{\boldsymbol{h}}_{\mathcal{C}_{\hat{y}}}^{l,h})\right)_{\hat{y}} \right]. \tag{16}$$

Intuitively, if the attention head is causal for class $\hat{y}$, this replacement will further reinforce the incorrect prediction, resulting in a small $U_C^{l,h}$. Conversely, if the head is misleading, the replacement will have a limited effect, leading to a larger $U_C^{l,h}$ than that of causal attention head.

**Utility D.** Finally, Utility D evaluates the effect of replacing attention head features in correctly classified samples of class $\hat{y}$, i.e., $\mathcal{C}_{\hat{y}}$, with the average from misclassified samples $\bar{\boldsymbol{h}}_{\mathcal{W}_{y,\hat{y}}}^{l,h}$:

$$U_D^{l,h} = \mathbb{E}_{\boldsymbol{x} \in \mathcal{C}_{\hat{y}}} \left[ f\left(\tilde{\Phi}(\boldsymbol{x}, \bar{\boldsymbol{h}}_{\mathcal{W}_{y,\hat{y}}}^{l,h})\right)_{\hat{y}} - f\left(\tilde{\Phi}(\boldsymbol{x}, \bar{\boldsymbol{h}}_{\mathcal{W}_{y,\hat{y}}}^{l,h})\right)_y \right]. \tag{17}$$

Since the data in $\mathcal{W}_{y,\hat{y}}$ is misclassified, the attention head features associated with the incorrect class $\hat{y}$, which are spurious features, tend to dominate the image representation, while the contribution from causal features relevant to the true class $y$ remains weak. Consequently, replacing the misleading heads (those aligned with $\hat{y}$) further increases the model's prediction confidence for class $\hat{y}$, whereas replacing the causal heads (those aligned with $y$) enhances the prediction for the correct class $y$. As a result, the misleading heads exhibit large values of $U_D^{l,h}$, while the causal heads correspond to smaller $U_D^{l,h}$ values.

**Summary of Utility Scores.** We propose four utility scores to analyze the role of attention heads. The first two, $U_A^{l,h}$ and $U_B^{l,h}$, compare samples with the same ground-truth label but different predictions. These scores help identify attention heads that contribute to inconsistent or unstable predictions. In contrast, $U_C^{l,h}$ and $U_D^{l,h}$ compare samples with different ground-truth labels but identical predictions, highlighting attention heads that may cause the model to confuse distinct classes.

Figure 6 illustrates these concepts using the Waterbird dataset [25], where the goal is to classify images as waterbirds or landbirds. In this dataset, the background (water or land) is a common source of spurious correlation. Since the four utility scores are designed to capture different types of features, but their specific behavior is difficult to validate in general, we leverage this known spurious feature to test them. By selecting failed samples in which background plays a key role, we can examine whether the scores respond as expected.

Specifically, using $U_A^{l,h}$ and $U_B^{l,h}$, we examine two types of samples: one correctly predicted as a waterbird with a water background, and another incorrectly predicted as a landbird, despite being a waterbird with a land background. If an attention head focuses on the background, it may help in the first case but harm in the second, indicating unstable behavior. $U_A^{l,h}$ and $U_B^{l,h}$ are proposed to identify these type of attention heads.

On the other hand, $U_C^{l,h}$ and $U_D^{l,h}$ focus on cases where the model predicts the same label for two semantically different samples, such as a waterbird on land and a landbird on water, both predicted as landbirds. Attention heads that rely on background features can mislead the model in both cases, suggesting they encode spurious correlations that negatively affect generalization. $U_C^{l,h}$ and $U_D^{l,h}$ are proposed to identify these attention heads.

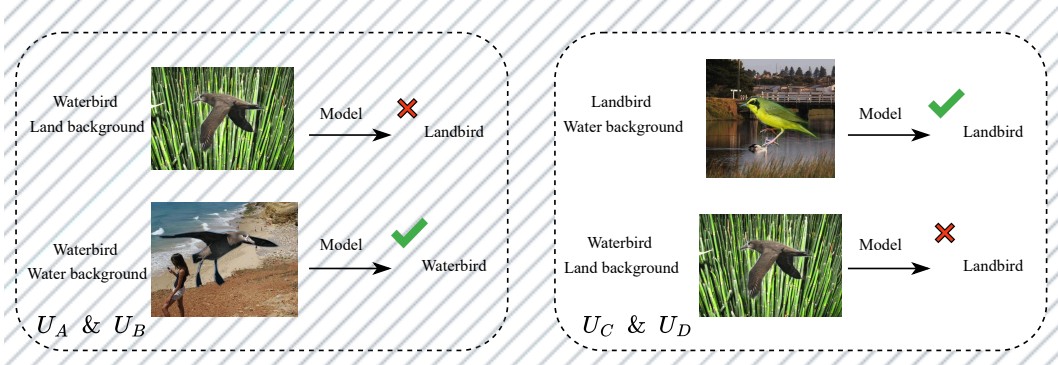

Figure 6: **Illustration of utility scores on the Waterbird dataset.** The computation of the four utility scores does not require carefully selected samples or prior domain knowledge. The examples shown here are chosen solely to illustrate the properties of the scores.

**Ranking and Ensembling of Utilities.** Each utility score ($U_A^{l,h}$, $U_B^{l,h}$, $U_C^{l,h}$, $U_D^{l,h}$) produces a ranked list of attention heads. For each utility, we select the top $T$ heads, and for each $t$ from 1 to $T$, we define $\mathcal{S}_t$ as the set of the top $t$ heads. To determine the optimal set of heads to ablate, we evaluate the validation utility of each candidate set $\mathcal{S}_t$ as follows:

$$\mathrm{U}(\mathcal{S}_t) = \mathbb{E}_{\boldsymbol{x}\in\mathcal{D}_A}\left[ f\Big(\Phi(\boldsymbol{x}) - \sum_{\boldsymbol{h}^{l,h}\in\mathcal{S}_t}\boldsymbol{h}^{l,h}(\boldsymbol{x})\Big)_{y_{gt}} - \sum_{y'\neq y_{gt}} f\Big(\Phi(\boldsymbol{x}) - \sum_{\boldsymbol{h}^{l,h}\in\mathcal{S}_t}\boldsymbol{h}^{l,h}(\boldsymbol{x})\Big)_{y'} \right], \qquad (18)$$

where $\mathcal{D}_A$ denotes the data set consisting of all available samples and $y_{gt}$ refers to the ground truth label for each corresponding image. The candidate set $\mathcal{S}_{\mathrm{ablate}}$ with the highest validation utility is selected for ablation.

**Summary of the Procedure.** For model editing tasks with multiple available samples, the first stage of RefineViT identifies the attention heads most responsible for prediction failures through the following systematic process:

1. **Score Calculation:** For each attention head, compute all four utility scores ($U_A^{l,h}$, $U_B^{l,h}$, $U_C^{l,h}$, $U_D^{l,h}$) as permitted by the available data.

2. **Candidate List Generation:** For each utility, rank the attention heads in descending order and select the top $T$ heads to form a candidate list.

3. **Ablation Candidate Generation:** For each candidate list, iteratively select the top $t$ heads (for $t = 1$ to $T$) as ablation candidates, denoted as $\mathcal{S}_t$.

4. **Validation Utility Evaluation:** For each ablation set $\mathcal{S}_t$, evaluate the validation utility (Eq. 18) on all available samples. The set $\mathcal{S}_t$ with the highest validation utility is chosen as the final set of attention heads for ablation, i.e., $\mathcal{S}_{\text{ablate}}$.

This strategy enables RefineViT to systematically identify and ablate the attention heads that most contribute to prediction errors, leveraging all available data for effective model editing.

# B    Experiment Details and Extra Results

## B.1    More Details in Section 6.1

The benchmark introduced by [32] comprises 16 types of misclassification for natural images and 22 types of misclassification for AI-generated oil painting images on ViT-B/16. Each sample serves as a single reference point for model editing, simulating a scenario where only one misclassified sample is available for correction. Other samples exhibiting the same type of prediction error as the reference sample are treated as neighboring samples within the editing scope for the generalization rate (GR) computation. Furthermore, 2,071 carefully curated images near the decision boundary of ViT-B/16, sourced from the validation sets of ImageNet-1K [3], ImageNet-R [13], and ImageNet-Sketch [30], are used as unrelated samples outside the editing scope for the calculation of the locality rate (LR).

For RefineViT, we fix the number of training epochs to 50 and use the Adam optimizer with a learning rate of 0.00002. For simplicity, the hyperparameter $\beta$ is set to 0. We evaluate the method using $\alpha \in \{10, 50, 100\}$. For each value of $\alpha$, we apply our approach to each sample in the benchmark—treating it as the sole available failed sample—and report the average performance across three evaluation metrics.

## B.2    More Details in Section 6.2: Performance Enhancement and Debiasing Scenarios

We randomly select 10 samples per class—waterbirds and landbirds for the Waterbirds dataset [25], and young and old celebrities for the CelebA dataset [19]—including both correctly and incorrectly predicted instances from CLIP-ViT. These samples are categorized into four groups based on their ground-truth and predicted labels. We then apply the four utility scores described in Appendix A, skipping any scores that are infeasible due to insufficient data in the comparison set. If the comparison reference set lacks sufficient data, we fall back to zero ablation rather than performing mean ablation.

Next, for each valid utility score, we obtain the top $T = 15$ attention heads, as a candidate list. Then we compute the validation utility score defined in Eq. (18), repeat for all candidate lists and select the attention head set with largest utility validation score as the final list.

For our method enhanced with a validation set, we perform ablation on the top $t$ ($t = 1, 2..., T$) heads in each ordered candidate list and select the one that delivers the best performance on the validation set, rather than relying on the validation utility score estimation.

Both our method and Tip-Adapter are evaluated in a zero-shot setting, leveraging CLIP-ViT's zero-shot capabilities without training additional classifiers. We show the results in Table 1 and Table 2.

**Stability Analysis**  To evaluate the stability of RefineViT, we run experiments on the Binary Waterbirds dataset with $n = 10, 20$, and $30$ samples per class, using four random seeds for each setting. Table 3 reports the mean and standard deviation of accuracies. RefineViT consistently demonstrates robust performance improvements across varying sample sizes and different random initializations, indicating its reliability and stability in data-scarce scenarios.

## B.3    More Details in Section 6.2: Model Editing Scenarios

**Datasets and Settings**  ImageNet-R [13] contains a diverse set of real-world images that CLIP-ViT can typically classify with high accuracy. We select the 18 classes with the most cartoon-style images

Table 2: Average-group accuracy (%) and worst-group accuracy (%) for classifying 'Young' or 'Old' on the CelebA dataset using CLIP-ViT-B/16.

| Method | Avg. ($\uparrow$) | Wst. ($\uparrow$) |
|---|---|---|
| Base | 70.1 | 43.8 |
| Tip-Adapter | 73.7 | 52.2 |
| Ours | **74.1** | **56.4** |

Table 3: Stability Analysis Results Across Varying Sample Sizes on the Binary Waterbirds Dataset

| Method | ViT-B/16 | | ViT-L/14 | | ViT-H/14 | |
|---|---|---|---|---|---|---|
| | Avg. ($\uparrow$) | Wst.($\uparrow$) | Avg. ($\uparrow$) | Wst. ($\uparrow$) | Avg.($\uparrow$) | Wst. ($\uparrow$) |
| Base | 72.8 | 45.6 | 75.5 | 47.7 | 68.6 | 37.2 |
| Ours (n = 10) | $79.8 \pm 1.2$ | $59.79 \pm 1.7$ | $85.5 \pm 0.8$ | $71.9 \pm 0.8$ | $74.5 \pm 1.0$ | $46.5 \pm 3.3$ |
| Ours (n = 20) | $80.8 \pm 0.2$ | $68.5 \pm 2.3$ | $85.7 \pm 0.4$ | $73.9 \pm 1.1$ | $74.9 \pm 0.6$ | $48.1 \pm 1.0$ |
| Ours (n = 30) | $80.4 \pm 0.2$ | $65.8 \pm 2.3$ | $85.5 \pm 0.4$ | $72.4 \pm 2.1$ | $73.5 \pm 1.7$ | $47.7 \pm 3.2$ |

from ImageNet-R and combine them with the Binary Waterbirds dataset [25], resulting in a new dataset with 20 classes. Our objective is to improve performance on the bird classes while preserving locality—minimizing unintended side effects on the ImageNet-R classes—using only 10 samples per bird class.

Similarly, we also evaluate RefineViT on ImageNet-A [14], a dataset of real-world images that are commonly misclassified by ResNet models. We select the 10 most populous classes and target the two worst-performing ones for improvement, while minimizing degradation on the remaining eight. This is done using only 4 samples from each of the two target classes.

**Implementation Detail** All experiments use the Adam optimizer with a fixed learning rate of $2 \times 10^{-5}$. We set both hyperparameters $\alpha$ and $\beta$ to 1000. For the standard fine-tuning baseline, we freeze the ViT backbone and train only an appended projection matrix—mirroring the architecture of our proposed RefineViT. All experiments are repeated with three random seeds to reduce variance due to random initialization.

## C   Experimental validation of attention heads identified

To validate whether RefineViT can effectively identify attention-head-level features responsible for prediction failures in the first stage, we test our four mean-ablation-based utility scores on the Binary Waterbirds Dataset [25]. This dataset combines thousands of waterbird and landbird images from the CUB dataset [29] with water or land backgrounds from the Places dataset [36]. Since the classification task focuses on bird type, the background introduces a significant known spurious correlation that can lead to prediction failures.

The core idea of our evaluation is that, while each utility score captures different aspects of attention heads' behavior in prediction failures, we design them to specifically identify heads that focus on spurious background cues. This is achieved by selecting samples in which the background is expected to be most prominent source of spurious correlation. By demonstrating that the attention regions identified by these methods are largely consistent, we validate that the scores behave as intended. This consistency is expected, as background-associated attention regions are fixed and should be detectable by all score variants.

Specifically, for $U_A^{l,h}$ and $U_B^{l,h}$, we use images where the background aligns with the bird type in correctly predicted samples and mismatches it in misclassified ones. For $U_C^{l,h}$ and $U_D^{l,h}$, to ensure that the background consistently contributes to prediction errors, we include only samples with mismatched backgrounds in both sets. An example of this selection is shown in Fig. 6.

We select 5 samples in each set, resulting in a total of 30 samples, with different correctly predicted samples used across the groups. We evaluate our method on the CLIP-ViT-L/14 model [18] and select the top $T = 15$ attention heads that induce the largest expected model shift for each utility score. The selected attention heads for each score are reported in Table 4, while the jointly selected attention heads and their corresponding TextSpan-generated[5] textual descriptions are presented in Table 5.

The results, presented in Table 4, show that $U_A^{l,h}$ and $U_B^{l,h}$, as well as $U_C^{l,h}$ and $U_D^{l,h}$, produce identical outcomes. This is expected, as each pair compares the same sets of samples and is based on the same underlying principle, leading to approximately the same results.

Notably, 8 of the 15 selected attention heads are shared across four utility scores. We present these shared attention heads along with their corresponding TextSpan-generated descriptions in Table 5. Furthermore, we use Grad-CAM [27] to visualize the regions these attention heads focus on, as shown in Figure 7.

The TextSpan-generated textual descriptions and Grad-CAM visualizations consistently show that the shared attention heads predominantly focus on background features. This confirms that the utility scores in RefineViT-stage-one effectively identify attention heads associated with prediction failures, thereby addressing **Q3**.

Table 4: Attention heads identified by each utility score. Those jointly selected attention heads are highlighted in bold. (**L22**,**H0**) represents the (**0**+1)-th head in the (**22**+1)-th layer.

| Utility A | Utility B | Utility C | Utility D |
|---|---|---|---|
| **(L23, H2)** | **(L23, H2)** | **(L23, H2)** | **(L23, H2)** |
| **(L23, H5)** | **(L23, H5)** | (L23, H6) | (L23, H6) |
| (L22, H6) | (L22, H6) | **(L22, H1)** | **(L22, H1)** |
| (L22, H4) | (L22, H4) | **(L23, H5)** | **(L23, H5)** |
| (L23, H14) | (L23, H14) | **(L23, H8)** | **(L23, H8)** |
| **(L23, H3)** | **(L23, H3)** | (L23, H0) | (L23, H0) |
| (L22, H2) | (L22, H2) | (L22, H5) | (L22, H5) |
| (L21, H0) | (L21, H0) | **(L23, H3)** | **(L23, H3)** |
| **(L23, H12)** | **(L23, H12)** | **(L23, H9)** | **(L23, H9)** |
| **(L23, H8)** | **(L23, H8)** | (L23, H1) | (L23, H1) |
| (L22, H12) | (L22, H12) | (L21, H9) | (L21, H9) |
| **(L23, H9)** | **(L23, H9)** | (L22, H9) | (L22, H9) |
| **(L22, H1)** | **(L22, H1)** | **(L23, H12)** | **(L23, H12)** |
| **(L23, H6)** | **(L23, H6)** | (L20, H10) | (L20, H10) |
| (L21, H15) | (L21, H15) | (L22, H0) | (L22, H0) |

## D  Ablation Study and Sensitivity Analysis

### D.1  Sensitivity Analysis of hyper-parameter T in Stage One

For model editing tasks with multiple available samples, the hyper-parameter $T$ in RefineViT stage one does not directly determine the number of attention heads to be ablated; rather, it defines the size of the pool from which we select attention heads based on their utility scores. Consequently, the performance improvement from ablation tends to plateau once $T$ is sufficiently large. To support this claim, we conduct a sensitivity analysis on the Waterbirds dataset [25] using both CLIP-ViT-B/16 and CLIP-ViT-L/14. Each experiment is repeated with three random seeds to reduce variability. As shown in Fig. 8, performance gains stabilize when $T$ exceeds 10.

### D.2  Ablation Study of Attention Heads Identified in Stage One

we conduct an ablation study to evaluate the contribution of the whole stage one in RefineViT. Specifically, we remove the influence of stage one by treating all available samples equally during stage two. In this setting, the model is trained to minimize the MSE with the initial model across

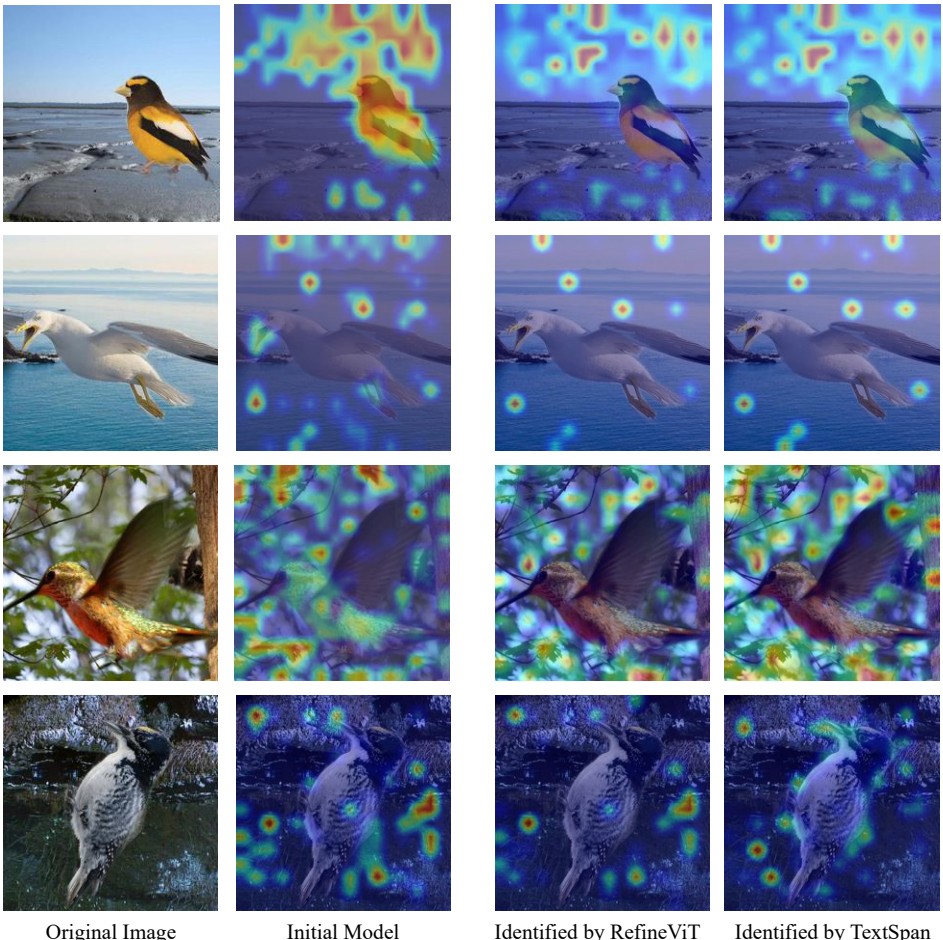

| Original Image | Initial Model | Identified by RefineViT | Identified by TextSpan |

Figure 7: **Grad-Cam visualization.** We present four examples from the Waterbirds dataset, each illustrating, from left to right: (1) the original image, (2) a heatmap showing the focus of the initial model, (3) a heatmap highlighting the attention heads identified by RefineViT as correlated with error sources based on all four utility scores, and (4) a heatmap highlighting the attention heads selected by TextSpan as related to the background. In the task of distinguishing waterbirds from landbirds, domain knowledge suggests that bird claws and beaks are causal features, whereas the background often introduces spurious correlations that lead to prediction errors. As shown in the examples, the attention heads identified by RefineViT predominantly focus on background regions. Moreover, compared to TextSpan, RefineViT selects attention heads that exhibit a stronger emphasis on the background and a weaker focus on causal features, despite TextSpan incorporating prior domain knowledge and requiring substantially more human effort and computational resources.

all available samples, rather than partitioning the samples into two groups—one aligned with the output of the ablated model and the other with that of the initial model. This variant is referred to as RefineViT (ablated). As shown in Fig. 9, the results demonstrate that stage one, where the attention heads responsible for errors are identified, effectively guides the error correction process for target classes. This guidance leads to smoother and more robust learning curves across different random seeds.

### D.3 Ablation Study of $\mathcal{L}_{\textbf{success}}(\theta)$ and $\mathcal{L}_{\textbf{locality}}(\theta)$

In accordance with Eq. (9), (10), and (12), the fine-tuning loss function is defined as a combination of the edit success loss, $\mathcal{L}_{\text{success}}(\theta)$, the edit locality loss, $\mathcal{L}_{\text{locality}}(\theta)$, and the cross-entropy loss, $\mathcal{L}_{\text{CE}}(\theta)$. To evaluate the contribution of the first two components to edit success and locality, we perform

Table 5: Common attention head with their top-5 results of TextSpan.

| Layer 23, Head 5 | Layer 23, Head 3 |
|---|---|
| Intertwined tree branches | Bustling city square |
| Flowing water bodies | Serene park setting |
| A meadow | Warm and cozy indoor scene |
| A smoky plume | Modern airport terminal |
| Blossoming springtime blooms | Remote hilltop hut |

| Layer 23, Head 8 | Layer 23, Head 6 |
|---|---|
| Photograph with a red color palette | Picture taken in Sumatra |
| An image with cold green tones | Picture taken in Alberta, Canada |
| Timeless black and white | Picture taken in the geographical location of Spain |
| Image with a yellow color | Image taken in New England |
| Photograph with a blue color palette | Photo captured in the Arizona desert |

| Layer 23, Head 2 | Layer 23, Head 12 |
|---|---|
| Image showing prairie grouse | Image with polka dot patterns |
| Image with a penguin | Striped design |
| A magnolia | Checkered design |
| An image with dogs | Artwork with pointillism technique |
| An image with cats | Photo taken in Galapagos Islands |

| Layer 23, Head 9 | Layer 22, Head 1 |
|---|---|
| ornate cathedral | A semicircular arch |
| detailed reptile close-up | An isosceles triangle |
| Image with a seagull | An oval |
| A clover | Rectangular object |
| Futuristic space exploration | A sphere |

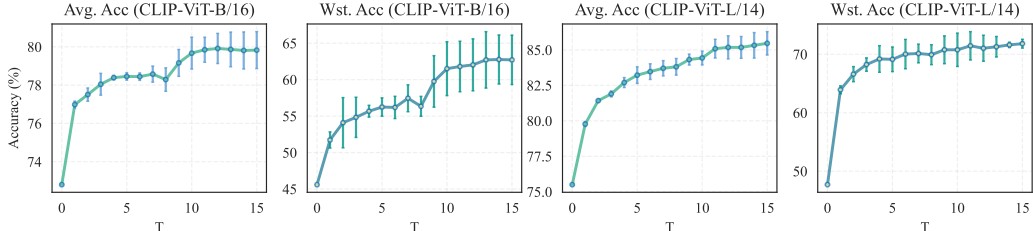

Figure 8: Sensitivity analysis for $T$

an ablation study on the 'Waterbirds + Imagenet-R' dataset by isolating each loss term. For clarity, the weight of the cross-entropy loss is fixed at 1, while the weights of $\mathcal{L}_{\text{success}}(\theta)$ and $\mathcal{L}_{\text{locality}}(\theta)$ are donated as $\alpha$ and $\beta$, respectively. All of our experiments are based on CLIP-ViT-L/14.

The ablation study for $\mathcal{L}_{\text{locality}}(\theta)$ is performed by fixing the weights of $\mathcal{L}_{\text{CE}}(\theta)$ and $\mathcal{L}_{\text{success}}(\theta)$ to 1 and 0, respectively, while varying the weight $\beta$ of $\mathcal{L}_{\text{locality}}(\theta)$ over the set $\{0, 10, 100, 1000\}$. Each hyperparameter configuration is evaluated using 20 samples selected under three fixed random seeds.

As shown in Fig.10, increasing $\beta$ generally leads to improvements in the average accuracies of both unrelated and target classes. Moreover, the variance across the three seeds diminishes with larger $\beta$ values, suggesting a more stable and robust training process. These results indicate that an appropriately chosen $\beta$ serves as an effective regularizer, guiding the model toward more consistent and generalizable representations.

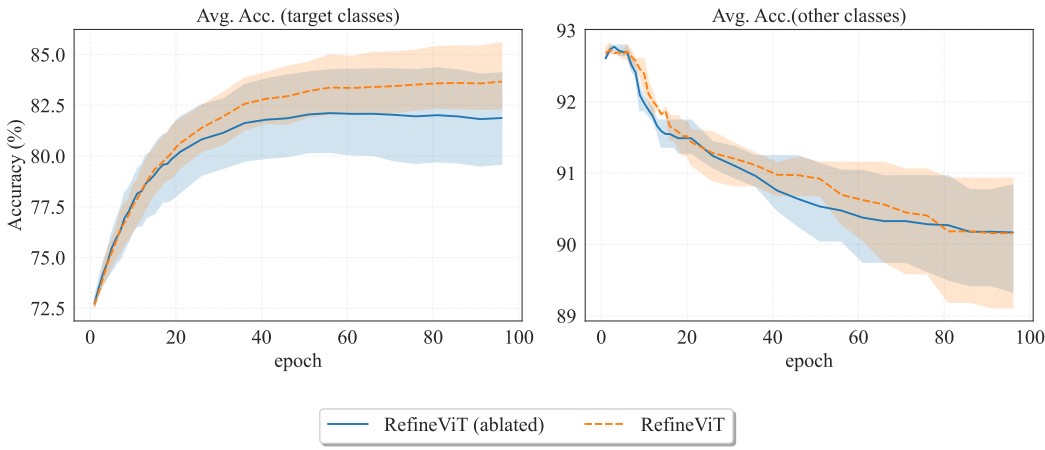

Figure 9: Performance comparison for the ablation study. The hyperparameters of our method are set to $\alpha = \beta = 1000$. For a fair comparison, the weight of the MSE loss in RefineViT (ablated) is also set to 1000.

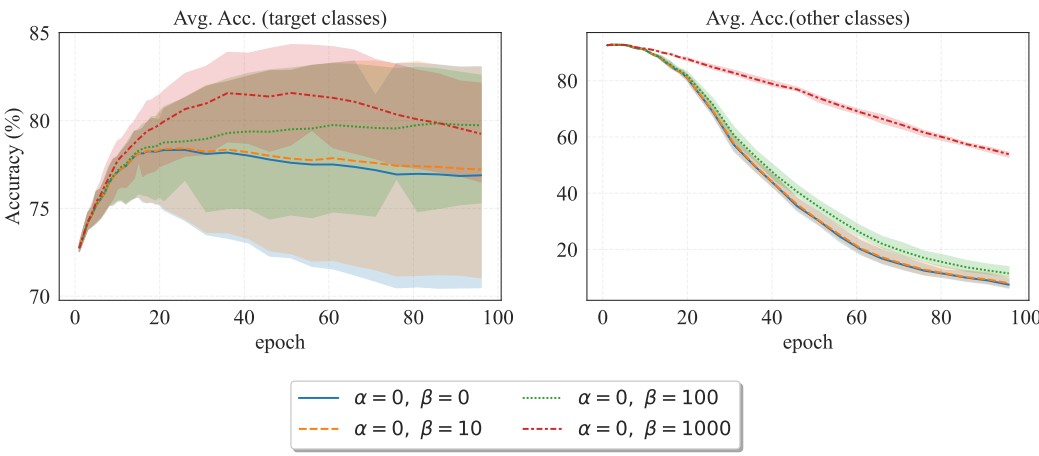

Figure 10: Ablation study for $\mathcal{L}_{\text{locality}}(\theta)$.

Similarly, we perform an ablation study on $\mathcal{L}_{\text{success}}(\theta)$ by varying its weight $\alpha$ over the set $\{0, 10, 100, 1000\}$, while keeping the weights of $\mathcal{L}_{\text{CE}}(\theta)$ and $\mathcal{L}_{\text{locality}}(\theta)$ fixed at 1 and 0, respectively. As shown in Fig. 11, increasing $\alpha$ yields notable gains in the average accuracy of target classes, accompanied by reduced variance across the three random seeds. In addition, the accuracy of unrelated classes also improves. This suggests that, although the success loss $\mathcal{L}_{\text{success}}(\theta)$ may conflict with the original locality objective $\mathcal{L}_{\text{locality}}(\theta)$, it helps mitigate overfitting, providing a more generalizable alternative to standard fine-tuning.

### D.4 Sensitivity Analysis of $\alpha$ and $\beta$

Based on the ablation study of $\mathcal{L}_{\text{success}}(\theta)$ and $\mathcal{L}_{\text{locality}}(\theta)$ conducted on the 'Waterbirds+ImageNet-R' dataset using CLIP-ViT-L/14, we observe that when the weights for $\mathcal{L}_{\text{success}}(\theta)$ and $\mathcal{L}_{\text{locality}}(\theta)$ are set to approximately $10^3$, our method achieves optimal performance in terms of both edit success and edit locality. To assess whether this weight range consistently yields stable performance across different datasets on CLIP-ViT-L/14, we performed a sensitivity analysis of the coefficients $\alpha$ and $\beta$ on 'Waterbirds+ImageNet-R' and 'ImageNet-A'. Specifically, we fixed the training epochs to 80 for 'Waterbirds+ImageNet-R' and 20 for 'ImageNet-A'. We used 3 fixed random seeds and computed

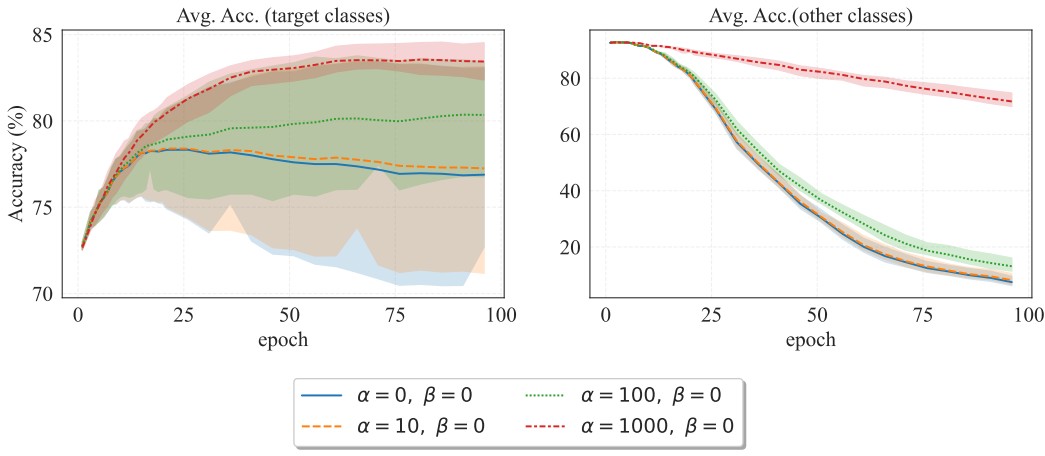

Figure 11: Ablation study for $\mathcal{L}_{\text{success}}(\theta)$.

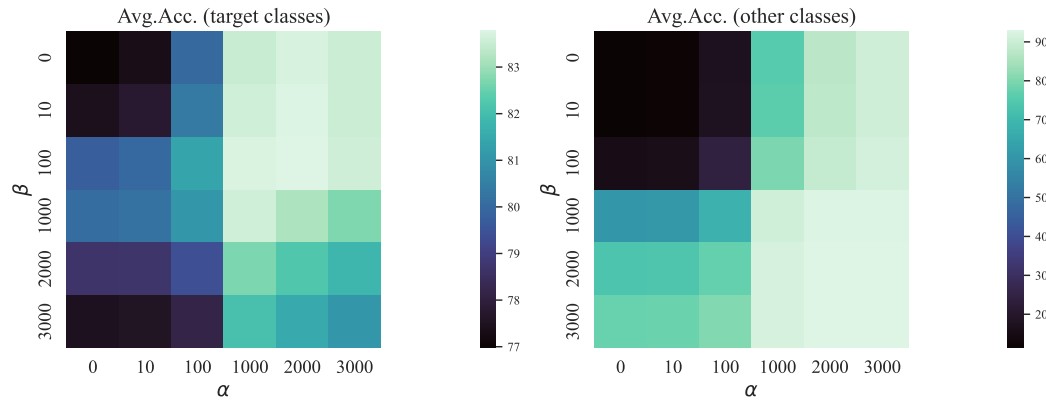

Figure 12: Sensitivity analysis on 'Waterbirds + ImageNet-R'

the average accuracy (in percentage) on the target classes and the unseen unrelated classes for each combination of $\alpha$ and $\beta$.

As illustrated in Figures 12 and 13, we observe a consistent improvement in accuracy on both the target and unrelated classes as $\alpha$ and $\beta$ increase. Notably, the accuracy on the target classes reaches its peak when $\alpha$ is approximately 1000 across both datasets. Furthermore, our results reveal a trade-off between $\mathcal{L}_{\text{success}}(\theta)$ and $\mathcal{L}_{\text{locality}}(\theta)$. Specifically, when the coefficient $\alpha$ for $\mathcal{L}_{\text{success}}(\theta)$ exceeds 1000, increasing the coefficient for $\mathcal{L}_{\text{locality}}(\theta)$ results in a degradation of accuracy on the target classes. This phenomenon is observed in both datasets.

### D.5 Qualitative Analysis of Individual Utility Score Effectiveness

In all experiments, we observe that when selecting the final list of ablated attention heads based on their performance on available samples, Utility C and Utility D generally outperform Utility A and Utility B in identifying heads for ablation. Notably, on the CelebA [19] dataset, ablating only the heads selected by Utility A and Utility B can significantly degrade performance, often leading the model to predict all samples as belonging to a single class. This failure arises from the implicit assumption behind Utility A and Utility B that the model is capable of fair prediction without systemic bias.

In practice, however, models often rely heavily on a narrow set of features and can exhibit strong bias toward dominant classes. For instance, when predicting whether a celebrity is young or old in CelebA [19], the CLIP-ViT-B/16 model tends to predict most individuals as "young". Although

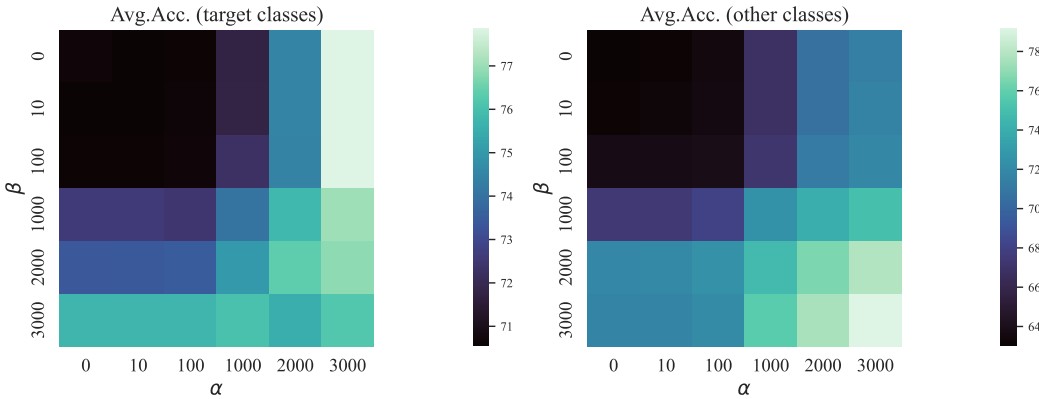

Figure 13: Sensitivity analysis on ImageNet-A

this yields high accuracy for genuinely young samples, it results in substantially lower accuracy for older individuals. A plausible explanation is that the model relies on simplistic visual cues—such as the presence of wrinkles—to make predictions: if wrinkles are detected, the model predicts "old"; otherwise, "young." Although this heuristic may be partially effective, it is insufficient, as not all older individuals exhibit visible wrinkles.

Utility A and Utility B tend to identify attention heads associated with features that are important but exhibit inconsistent behavior across correctly and incorrectly classified samples. In the case of CelebA, this can lead to the unintended selection of heads tied to wrinkle-related features, which are mistakenly flagged as harmful due to their differing influence on the model's predictions across these groups. As a result, ablating these heads removes features that are essential but not always correct, which further amplifies the model's bias and ultimately leads it to predict a single class for all inputs.

In contrast, Utility C and Utility D aim to identify attention heads that negatively impact both correctly and incorrectly classified samples. This leads to more conservative and balanced modifications, making them more effective in challenging cases like CelebA.

### D.6 Update Strategies

In Section 3.2, we demonstrate that spurious correlations are a key factor contributing to prediction failures in computer vision tasks. To mitigate the influence of misleading features in the learned representations, RefineViT employs a trainable projection matrix after the transformer blocks as an update strategy, rather than directly fine-tuning the later MSA modules, which may distort the extracted features.

To further validate the effectiveness of this design choice, we compare it with LoRA, a widely used update strategy for MSA modules. Specifically, we apply LoRA to the last two MSA layers during the second stage of RefineViT, as these later layers have been shown to play a crucial role. Due to its significantly higher computational cost compared to the original RefineViT, this experiment is conducted only on one group from the Natural ViT Benchmark [32].

The rank in LoRA serves as a hyperparameter that balances fine-tuning capacity and the risk of overfitting by controlling the number of trainable parameters. In our hyperparameter search, we find that values of $\alpha \in [0, 1]$ generally yield better performance. Accordingly, for each rank, we evaluate LoRA with $\alpha \in \{0, 0.1, 0.5\}$ and compare the results to the original RefineViT using the projection matrix. As shown in Figure 14, using LoRA in the last two MSA layers results in substantially worse performance, despite requiring much greater computational resources. These observations further support the effectiveness of our proposed update strategy.

### D.7 Extension to Segmentation Task

To examine the adaptability of our method beyond image classification, we conduct a preliminary experiment on a CLIP-based segmentation task following the setup in [5]. In contrast to classification,

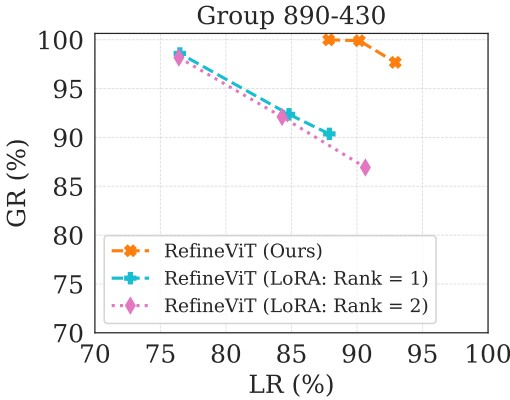

Figure 14: Comparison of update strategies: LoRA vs. projection matrix

segmentation requires the model to process all patch tokens to predict pixel-level outputs, which introduces additional challenges for attribution and decomposition during model editing. Using a single edit sample, we apply our method to the segmentation model and evaluate the resulting performance in terms of pixel-wise accuracy.

The edited model achieves higher accuracy on images similar to the edit sample and a slight improvement in overall performance, as summarized below:

| Model | Similar Data | Overall |
|---|---|---|
| Original CLIP-ViT | 72.29% | 76.78% |
| CLIP-ViT-Edited (Ours) | **74.50%** | **77.01%** |

These results suggest that our framework can be adapted to more complex vision tasks such as segmentation, showing its potential generality beyond classification.

