# OpenReview forum: "Model Editing for Vision Transformers"
_NeurIPS.cc/2025/Conference — NeurIPS 2025 poster_

### Official Review · Reviewer_yWzz · 2025-06-30

**Clarity:** 4
**Significance:** 3
**Originality:** 3
**Rating:** 5
**Confidence:** 4

**Summary:**

This paper presents RefineViT, a novel two-stage framework for model editing in Vision Transformers (ViTs). The authors make an observation that, unlike Language Models, ViT predictions are more sensitive to perturbations in the Multi-Head Self-Attention (MSA) layers than the MLP layers. Capitalizing on this, RefineViT identifies and ablates erroneous attention heads to correct misclassifications. To preserve performance on unrelated inputs, the method learns a projection matrix that aligns the edited representation with the original, unperturbed one. The experiments demonstrate that this MSA-centric approach is more effective for ViTs than directly adapting LM editing techniques.

**Questions:**

* The paper's central claim is that MSA layers are more critical than MLP layers for storing factual knowledge in ViTs. Could the authors provide more intuition or a theoretical argument for why this might be the case?
* If you repeat Stage 1 using two different misclassified examples of the same class, how consistent is the top-T list?
* Figure 4: it looks like ROME, which is a method thats put strong constraints on MLP, performs really well on locaility. It seems to be a little contradicting to the core finding of this paper (MSA plays a more important role in preserving the model's capability).
* L202: "This allows updates in under 0.3 seconds for 50 epochs". Does that include the time to find the ablation matrix A?

**Ethical Concerns:**

["NO or VERY MINOR ethics concerns only"]

**Final Justification:**

I would like to thank the authors for providing detailed and well-structured responses. The rebuttal addresses my concerns and I will keep my rating.

**Limitations:**

Yes

**Quality:**

4

**Strengths And Weaknesses:**

**Strengths:**

Fundamental Insight - The core contribution of this paper is the empirical finding that MSA layers are the primary locus of factual knowledge in ViTs. This is a significant departure from the established wisdom in the NLP domain, where MLP layers are considered the key-value memories. This insight alone is a valuable contribution to our understanding of vision transformers.

Methodological Soundness - The two-stage design of RefineViT follows logically from the empirical findings. The selective ablation of attention heads combined with projection learning is principled and effective.

**Weaknesses:**

Ablation Needed - There needs to be experiments to understand that which component in this approach is more critical. In multi-label settings, can you ablate the identified heads without learning P, and conversely train P without zeroing heads, to quantify each stage’s individual impact on GR–LR performance?

Computation Complexity - Computing head utilities requires O(L\!\times\!H) forward passes per sample. Should A be computed for each sample independently?

---

> ### Author Rebuttal · Authors · 2025-07-31
>
> We sincerely thank the reviewer for recognizing our insight that MSA layers are the primary knowledge locus in ViTs, as well as the principled and effective two‑stage design of RefineViT. Below, we address all weaknesses and questions, supplemented with new experiments for further support.
>
> **W1**. Ablation studies to clarify the individual impact of each component: 1.ablate without learning P in multi-label setting. 2. training P without zeroing heads
> > We conducted two ablation experiments to assess the individual impact of the two stages:
> >
> > 1. **Ablation without learning P**: We disable Stage Two and directly ablate the identified heads. On the ViT Natural Image benchmark (class 954\_582), this results in a **sharp drop in Locality Rate (LR)** as more heads are ablated, confirming that learning P is critical for preserving locality (see table below).
> >
> > | N\_ablated | 0   | 3           | 6          | 9          | 12          |
> > | ---------- | --- | ----------- | ---------- | ---------- | ----------- |
> > | LR(%)      | 100 | 46.16±2.09  | 38.08±4.91 | 32.94±8.55 | 29.03±10.37 |
> > | GR(%)      | 0   | 73.78±13.08 | 86.67±2.30 | 84.89±2.03 | 84.44±2.78  |
> >
> > 2. **Training P without zeroing heads**: This is already shown in Figure 9 (Appendix, Page 20). Without Stage One (head zeroing), the GR gains largely vanish, highlighting its role in guiding P's learning.
> >
> > Together, these results demonstrate that both stages are essential and complementary.
>
>
> **W2**.Computation Complexity: $\mathcal{O}(L! \times H)$ forward passes per sample?
> > In practice, we perform **only one forward pass per sample**. The required components for computing head utilities (i.e., the decomposition of the backbone output) are stored and reused. This design avoids the $\mathcal{O}(L! \times H)$ cost and keeps our method computationally efficient. A detailed description of the head‑selection procedure is provided in Appendix (p.14, Summary of the Procedure).
>
> **Q1**.More intuition or theoretical argument why MSA are more critical?
>
> > The intuition is that in ViT architectures, the lower layers primarily focus on extracting local features from individual patches (e.g., parts of an object like a wing or a head), while higher layers integrate these local features into composite concepts (e.g., recognizing a bird as a combination of head, wings, claws, and tail).
> >
> >Since MSA layers capture relationships between patches, errors in these layers can significantly disrupt the formation of composite concepts, leading to a larger impact on prediction accuracy. In contrast, MLP layers mainly refine the information extracted by MSA but do not directly capture inter‑patch relationships, making their impact comparatively less critical.
>
> **Q2**. If repeating stage 1 using different samples, how consistent are the resulting top-K list?
>
> > We conduct an experiment where we randomly selected two images from the ViT editing benchmark (class 954_582) and measured the overlap in their top-K head selections. We repeated this process 100 times and averaged the results to minimize randomness. The results show that the top-K lists have an average overlap of over 65%, indicating good consistency. Specifically, the overlap percentages for different values of K are as follows:
> >
> > | Top-K | K=1 | K=3 | K=6 | K=9 | K=12 | K=15 |
> > |-------|-----|-----|-----|-----|------|------|
> > | Overlap (%) | 73.0 | 81.33 | 68.67 | 65.11 | 66.92 | 69.87 |
> >
> > These findings demonstrate that our method produces consistent top-T lists even when using different misclassified examples from the same class.
>
> **Q3.** Why ROME (update last layer of MLP) performs really well on locality? Does it contradicts the core finding of this paper (MSA plays a more important role in preserving the model's capability).
>
> > **No, it does not contradict.**
> > - Locality measures how well predictions are preserved on unrelated data. ROME edits only the MLP layer and leaves the MSA modules untouched, so the model's core representations are largely preserved, leading to a higher Locality Rate (LR).
> > - However, its high locality can come at the cost of weak edits. As shown in Figure 4, ROME has a relatively low Generalization Rate (GR), indicating limited editing success. In fact, if no editing were performed, the Locality Rate would remain at 100%, highlighting the trade-off between locality and effective editing.
> >
> > Thus, our core insight remains valid: MSA modules are central to knowledge preservation and effective editing.
>
> **Q4.L202**: "This allows updates in under 0.3 seconds for 50 epochs". Does that include the time to find the ablation matrix A?
> > The reported update time of under 0.3 seconds for 50 epochs does not include the time to compute the ablation matrix A. However, as noted in our response to W2, Stage One requires only a single forward pass per sample, after which the backbone’s output is stored. Subsequent steps involve only matrix multiplications (for CLIP-ViT) or classifier-head-only forward passes, making them extremely fast with negligible time cost. Thus, the overall process remains highly efficient.

---

### Official Review · Reviewer_uxyq · 2025-07-01

**Clarity:** 3
**Significance:** 3
**Originality:** 3
**Rating:** 4
**Confidence:** 3

**Summary:**

This paper proposes RefineViT, a model editing method tailored for Vision Transformers (ViTs).   Unlike prior approaches that adapt language model (LM) editing strategies by modifying MLP layers, the authors show that ViT predictions are more influenced by multi-head self-attention (MSA) modules.  RefineViT identifies problematic attention heads responsible for prediction errors and selectively ablates or projects their outputs to correct mistakes while preserving performance on unrelated data.   The method is validated on several image classification benchmarks, showing superior correction and locality compared to prior ViT editing approaches.

**Questions:**

1. How does the approach perform if many edits are required, e.g., in dynamic or continuously evolving datasets?  Are there cumulative effects or degradation as the number of edits increases?

2. Can the proposed framework be adapted for more complex vision tasks such as object detection, instance/semantic segmentation, or video tasks?  What are the main challenges or required modifications?

**Ethical Concerns:**

["NO or VERY MINOR ethics concerns only"]

**Final Justification:**

The author’s response addressed my concerns. I have no further questions and will keep my original score. I look forward to seeing the author extend model editing to more complex tasks in the future.

**Limitations:**

See the weaknesses.

**Quality:**

3

**Strengths And Weaknesses:**

# Strengths

1. The paper studies the practical problem of model editing for ViTs, which may be important as models are increasingly used in real-world applications where errors can occur after deployment.

2. The main contribution is to target the MSA modules for editing, rather than the traditional approach of editing MLP layers.  This idea is supported by some experimental analysis suggesting that ViT predictions are more sensitive to attention modules.

3. The method RefineViT consists of two stages: identifying which attention heads contribute to incorrect predictions and then adjusting or projecting their outputs to correct the errors.  This process seems relatively lightweight, as it does not require retraining the entire model.

4. Experiments are carried out on several classification datasets, and results show improvement over some existing model editing techniques on these benchmarks.


# Weaknesses

1. All of the experiments and analyses are limited to image classification tasks. There is no experimental evidence for whether the approach works for other vision tasks such as object detection or segmentation.

2. Most experiments are conducted on standard or synthetic classification datasets, such as ImageNet-R and ImageNet-A, which may not fully reflect large-scale or highly complex real-world scenarios.

---

> ### Author Rebuttal · Authors · 2025-07-31
>
> We sincerely thank the reviewer for recognizing the practical importance, MSA‑focused design, and strong experimental results of our RefineViT framework. Below, we address all weaknesses and questions, supplemented with new experiments for further support.
>
> **Q1.** Does this method generalize to scenarios where multiplex edits are required? dynamic and continuously? cumulative effects or degradation?
> > Yes, our method supports multiple editing and demonstrates strong performance in such scenarios.
> > - We evaluate this on the ViT benchmark by sequentially editing three target classes, each involving a single misclassified example.
> >     - We compare our approach against the state-of-the-art method (WTE) and standard fine-tuning with $l_{2}$ regularization.
> >     - At each step, we measure both Generalization Rate (GR), i.e., accuracy on target classes, and Locality Rate (LR), i.e., performance on unrelated data. Importantly, we track GR for both the current (e.g., CLS_3 for the Editing 3) and all previously edited target classes (e.g., CLS_1 and CLS_2) to assess edit retention. The results are shown in the following tables.
> >
> > |GR|Edit 1|Edit 2 |Edit 2 | Edit 3 | Edit 3| Edit 3|
> > |--|----|---|--|-----|---|--|
> > ||CLS_1|CLS_1|CLS_2|CLS_1|CLS_2|CLS_3|
> > |RefineViT(ours)| 92.0| 92.0| 98.54|90.67|98.54|96.1|
> > |WTE|73.61| 73.61| 84.21 |73.61| 84.21 |96.54
> > |Std.F.T|38.89| 0| 37.84 |5.55| 5.26 |98.77
> >
> > |LR|Edit 1|Edit 2|Edit 3|
> > |--|----|---|--|
> > |RefineViT(ours)|91.79|88.12|84.69|
> > |WTE|90.48 |86.52 |85.176 |
> > |Std.F.T|95.99| 83.84| 83.88
> >
> > - Compared to the SOTA method (WTE) and standard fine-tuning with \$l\_2\$ regularization, our approach consistently maintains higher GR and LR across all edits. This indicates that **RefineViT better preserves previous edits while maintaining locality**, showcasing its effectiveness in multiple editing settings.
>
> **Q2 & W1.** Can the method be adapted for more complex vision tasks such as object detection, instance/semantic segmentation? What are the main challenges or required modifications?
> > Extending model editing to complex tasks like **object detection or segmentation** presents several challenges:
> > 1. These tasks involve **different error types** and outputs (e.g., bounding boxes or masks rather than class labels).
> > 2. Unlike classification, which primarily uses the **CLS token**, detection/segmentation rely on **all patch tokens**, making attribution and decomposition more difficult.
> > 3. There is currently **no established benchmark or protocol** for model editing in these settings, making standardized evaluation difficult.
> >
> > Despite these challenges, we demonstrate the **potential of our method beyond classification** by applying it to a **CLIP-based segmentation task**, following the setup in \[4]. Using a single edit sample, we observe improved pixel-wise accuracy on similar images and a slight improvement in overall performance:
> >
> > | Avg. Pixel-Wise Accuracy | Similar Data | Overall |
> > | ------------------------ | ------------ | ------- |
> > | Original CLIP-ViT        | 72.29%       | 76.78%  |
> > | CLIP-ViT-Edited (Ours)   | 74.50%       | 77.01%  |
> >
> > This shows initial promise for adapting our framework to segmentation tasks. We plan to explore this direction further and develop benchmarks and methods for model editing in segmentation.
> >
> >[4] Yossi Gandelsman, Alexei A. Efros, and Jacob Steinhardt. Interpreting clip’s image represen- tation via text-based decomposition.

---

### Official Review · Reviewer_x93k · 2025-07-01

**Clarity:** 3
**Significance:** 3
**Originality:** 3
**Rating:** 4
**Confidence:** 4

**Summary:**

This paper proposes a new method for editing Vision Transformers in classification tasks by localizing the self-attention heads most responsible for incorrect predictions. The authors then introduce a technique to neutralize the effect of these heads by applying a linear transformation to the embeddings—designed to minimally affect unrelated samples while producing outputs that approximate those obtained when the localized heads are ablated.

**Questions:**

Please read weaknesses.

**Ethical Concerns:**

["NO or VERY MINOR ethics concerns only"]

**Final Justification:**

I still think borderline accept could be a good score for this paper, as I see the application of the approach a bit limited, and also the core idea of localization explored in the literature.

**Limitations:**

yes

**Paper Formatting Concerns:**

Follows the guideline.

**Quality:**

3

**Strengths And Weaknesses:**

Strengths:
The paper is well written, and the idea of editing the model by localizing important MSA layers is clearly presented. The experiments are extensive, demonstrating improvements over strong baselines and highlighting the effectiveness of the proposed method. Figure 5 is particularly insightful, showcasing that a lightweight projection matrix applied to localized MSAs can be highly impactful.

Weaknesses:
Figures 3 and 4 could benefit from clearer and more informative visualizations. Additionally, it would strengthen the analysis to include a comparison where a random subset of MSA layers—rather than the localized ones—is used to optimize the projection matrix. This would help assess whether the identified layers are truly critical. Furthermore, the decomposition of residual tokens into contributions from MSA and MLP layers is not novel, as it has been explored in several prior works [1], limiting the novelty of this aspect.




-----
1. Balasubramanian, Sriram, Samyadeep Basu, and Soheil Feizi. "Decomposing and interpreting image representations via text in vits beyond CLIP." arXiv preprint arXiv:2406.01583 (2024).

---

> ### Author Rebuttal · Authors · 2025-07-31
>
> We sincerely thank the reviewer for recognizing our clear presentation, extensive experiments, and the effectiveness of the lightweight projection on localized MSA layers. Below, we address all weaknesses, supplemented with new experiments for further support.
>
> **w1.** Figures 3 and 4 could benefit from clearer and more informative visualizations.
> >**A:** Thanks for your suggestion. We will revise them.
>
> **W2.** Include a comparison where a random subset of MSA layers is used to optimize the projection matrix?
>
> > **A:** Thank you for the suggestion. We conducted this comparison using the Waterbirds dataset and ViT-L-14. The results are summarized below:
> >
> > |GR |Epoch 0|Epoch 20|Epoch 40| Epoch 60|Epoch 80|
> > |--|----|---|--|-----|---|
> > |RefineViT| 72.64| 80.62±1.20| 82.82±1.25|83.35±1.41|83.58±1.25|
> > |Random Subset|72.64|79.41±1.72| 81.65±1.56|82.47±1.81|82.59±2.09|
> >
> >
> > |LR |Epoch 0|Epoch 20|Epoch 40| Epoch 60|Epoch 80|
> > |--|---|---|--|-----|---|
> > |RefineViT|92.65|91.43±0.25|91.36±0.27|90.96±0.54|90.18±0.89
> > |Random Subset|92.65|91.32±0.17|90.59±0.16|90.97±0.47|90.62±0.46
> >
> > We find that randomly selecting MSA layers for projection matrix optimization leads to consistently lower GR  across training epochs compared to our structured selection, underscoring the importance of the identified heads for effective model editing.
>
>
> > Furthermore, **Figure 9 (page 20) presents additional ablations on stage 1 of RefineViT,**  further demonstrating the value of our head selection mechanism. Taken together, these results confirm that our design choices are essential for achieving strong generalization while preserving locality.
>
> **W3.** The decomposition of residual tokens into contributions from MSA and MLP layers is not novel.
>
> >**A:** We clarify that the decomposition of ViT representations is **not presented as a novel contribution** of our work. It appears in the **Preliminaries section** with proper citations, serving only to establish the foundation for our method. Our main contribution lies in the editing framework and targeted refinement strategy, which go beyond these prior analyses.
>
> > We will also include a citation to the paper you mentioned in the related work.

---

> > ### Comment · Reviewer_x93k · 2025-08-04
> >
> > Thanks for the rebuttal and the additional experiments. I observe only marginal improvements when using RefineViT compared to a random subset, which suggests that the core contribution of localization may be limited. In fact, it raises the question of whether localization is truly necessary as a first step. With these concerns, I would like to keep my score as is.

---

> ### Author Response · Authors · 2025-08-05
>
> We thank the reviewer for the thoughtful feedback. We would like to clarify the specific contribution of our attention head identification step.
>
> 1. **Attention Head Identification Improves Generalization Rate (GR) Without Fine-Tuning:**
>    As shown in the table below, ablating attention heads identified by our method leads to a significant increase in GR compared to both the base model and random subset ablation, without any fine-tuning. Specifically:
>
>    | Method       | GR(%)   |LR(%)   |
>    |--------------|-------- |--------|
>    | Base         | 72.64   | 92.65  |
>    | RefineViT    | 83.48   | 88.19  |
>    | Random Subset| 72.03 ± 0.72 | 90.98 ± 2.28|
>
>    Our method effectively removes attention heads that contributed to the error, resulting in improved GR. In contrast, random subset ablation performs similarly or slightly worse than the base model. The effectiveness of attention head identification is also supported by the empirical results in Table 1 of the paper.
>
> 2. **Fine-Tuning Narrows Performance Gap:**
>    After fine-tuning, the gap between attention head identification and the random subset becomes less pronounced. This is partly because fine-tuning with an appropriate loss can itself be a strong model editing method, efficiently leveraging the information from the editing sample, as also shown in [A]. Thus, even random subset ablation followed by fine-tuning achieves competitive results. Nevertheless, our method consistently outperforms random subset ablation due to the additional information provided by attention head identification.
>
>
> 3. **Necessity of Attention Head Identification:**
>    These findings suggest that attention head identification is particularly valuable without fine-tuning, providing consistent GR improvement. While fine-tuning can compensate for suboptimal ablation, the localization step remains important for achieving better results.
>
> We hope this clarifies the unique role and necessity of our attention head identification step.
>
> [A] Gangadhar, Govind, and Karl Stratos. "Model editing by standard fine-tuning." Findings of ACL 2024.

---

### Official Review · Reviewer_qQpH · 2025-07-03

**Clarity:** 3
**Significance:** 3
**Originality:** 3
**Rating:** 4
**Confidence:** 3

**Summary:**

This paper introduces a two-stage framework, in the context of model editing for Vision Transformers (ViTs), to correct prediction errors without requiring expensive retraining. The proposed method is called RefineViT. The authors first present an empirical analysis arguing that, unlike in language models where MLP layers are the primary target for editing, the Multi-Head Self-Attention modules are the dominant drivers of predictions in ViTs. Based on this observation, the first stage of RefineViT identifies specific attention heads responsible for a given misclassification by ablating their contributions and measuring the impact on the model's output logits. The second stage, termed "representation rectification," learns a lightweight projection matrix that modifies the ViT's final image representation. This matrix is trained to alter the representation of erroneous samples towards their corrected versions, while preserving the representations of correctly classified samples, thus balancing edit success with locality. The method is shown to achieve state-of-the-art performance on a ViT editing benchmark and demonstrates effectiveness in debiasing tasks for CLIP-ViT models.

**Questions:**

q1. The L_locality term is a central part of your proposed method for balancing success and locality. Yet, for the main results in Figure 4, it was disabled ($\beta=0$). Please provide a clear justification for this experimental choice.

q2. Your identification stage relies on a linear decomposition of the ViT representation (Eq. 4). Could you please discuss the potential impact of this simplification? Specifically, can you characterize scenarios where this approximation might fail, leading to the misidentification of faulty heads? Acknowledging this would strengthen the paper's scientific integrity.

q3. Does this method generalize to scenarios where multiplex edits are required?


I am open to increasing the score provided that the issues are reasonably addressed.

**Ethical Concerns:**

["NO or VERY MINOR ethics concerns only"]

**Final Justification:**

The authors provide more details to justify some of their design choices, which triggered my initial concerns. Based on this, I think these issues can be considered addressed , though these design choice are still the results of compromise to some extent. Thus I increase my score to 4 but not an even higher one.

**Limitations:**

The authors have discussed the limitations, but it is expected to be expanded, for example, the discussion of core methodological assumptions should be in the main text.

**Quality:**

3

**Strengths And Weaknesses:**

### Strengths

1. This paper finds that ViT editing should target attention modules instead of MLP modules. This is an interesting conceptual contribution, which challenges the direct transfer of paradigms from the language model editing literature and instead proposes a solution tailored to the specific architectural properties of ViTs. I think this could guide future studies, upon further verification.
2. The two-stage approach is reasonable and is connected to the identified gap. The use of a projection matrix for rectification helps to avoid invasive changes to the model's backbone and is computationally efficient, making it practical for targeted applications.
3. The method shows substantial improvements over existing methods on a benchmark. The hypothesis is further supported by ablation studies.

---

### Weaknesses

1. The entire method is built on a simplified linear decomposition of the ViT's representation (Eq. 4), which treats attention head contributions as independent and additive. While I understand that the linearity assumptions are widely adopted in the model editing literature, in my opinion, this is a significant simplification of the true, non-linear, and even sequential nature of the Transformer architecture. The paper does not acknowledge or analyze the limitations of this core assumption.
2. There is a discrepancy between the described method and the main experiments. The `L_locality` term is a key component of the proposed objective function (Eq. 12). But it was disabled (setting $\beta=0$) for the main benchmark results (according to Appendix B.1). Doesn’t this make the locality preservation effort invalid?
3. There is limited discussion of when the proposed method might fail or its assumptions might be violated.

---

> ### Author Rebuttal · Authors · 2025-07-31
>
> We sincerely thank the reviewer for recognizing our conceptual contribution, practical two‑stage design, and strong empirical results supported by ablations. Below, we address all questions and weaknesses, supplemented with new experiments for further support.
>
> **W1 & Q2.** The method relies on a linear decomposition of ViT representations. What are the limitations of this assumption, and in what cases might it lead to misidentification of faulty heads?
> >**A:** We acknowledge this simplification and discuss it in the Limitations section (page 23). Our decomposition focuses on the first‑order (direct) contributions of attention heads to the final representation, omitting higher‑order interactions across subsequent layers. While this does not fully capture the non‑linear and sequential dynamics of transformers, it provides a **tractable and effective** editing process.
> >
>
> > We understand the reviewer’s concern that some heads may contribute to incorrect predictions primarily through higher‑order effects, or vice versa. However, our method modifies the final representation by removing only the direct (first‑order) contributions of identified heads, leaving their indirect effects intact. Therefore, the approach is generally robust to potential misidentification caused by omitting higher‑order effects.
> >
>
> > Investigating higher‑order contributions is a promising future direction for achieving **more fine‑grained and accurate interventions**.
>
>
> **W2 & Q1.** The L_locality term is a central part of the method for balancing success and locality. Why was it disabled for main benchmark in Figure 4?
>
> >**A:** We clarify that $L_{\text{locality}}$ is indeed a key component for balancing edit success and locality. However, in the experiments shown in Figure 4, only one misclassified sample is available, and no correctly predicted samples are present. Under such conditions, the standard form of $L_{locality}$ (Eq. 10) is not applicable.
> >
>
> >Although an alternative regularization form (Eq. 11) could be used to reduce overfitting, we observed that our method already maintains strong locality without it. This aligns with our sensitivity analyses (Figures 10–13, Appendix), which shows that the success loss $L_{\text{success}}$ alone implicitly helps mitigate overfitting. Moreover, the small learning rates and few training epochs used in these experiments also help limit overfitting.
> >
>
> >In practice, we find the optimal $\beta$ is significantly smaller than $\alpha$ in these specific experiments. Given the minor performance gap and for simplicity, we set $\beta = 0$ in these experiments.
>
> **W3.** Limited discussion of when the proposed method might fail or its assumptions might be violated.
>
> >**A:** First, as we acknowledged in the Limitations section (Page 23), our method models **only the first‑order**, direct contributions of individual heads to the final representation and does not capture higher‑order or indirect effects; in scenarios where higher-order effects dominate, the proposed method cannot repair the corresponding errors.
> >
>
> > Second, in the single‑sample editing setting, our method depends heavily on the quality of the provided misclassified sample. If the sample is ambiguous or mislabeled, the effectiveness of the edit will degrade.
> >
>
> > We will include these discussions in the Limitation section.
>
> **Q3.** Does this method generalize to scenarios where multiplex edits are required?
>
> >**A:** Yes, our method supports multiplex editing and shows strong performance in such scenarios.
> >> - Evaluation setup: We test on the ViT benchmark by sequentially editing three target classes, each with a single misclassified sample.
> >>     - We compare our approach against the state-of-the-art method (WTE) and standard fine-tuning with $l_{2}$ regularization.
> >>     - At each step, we measure both Generalization Rate (GR), i.e. accuracy on target classes, and Locality Rate (LR), i.e. performance on unrelated data. **Importantly, we track GR for both the current (e.g., CLS_3 for the 3rd Edit 3) and all previously edited target classes (e.g., CLS_1 and CLS_2) to assess edit retention.** The results are shown in the following tables.
> >>
> >> |Generalization Rate (GR)|Edit 1|Edit 2 |Edit 2 | Edit 3 | Edit 3| Edit 3|
> >> |--|----|---|--|-----|---|--|
> >> ||CLS_1|CLS_1|CLS_2|CLS_1|CLS_2|CLS_3|
> >> |RefineViT(ours)| 92.0| 92.0| 98.54|90.67|98.54|96.1|
> >> |WTE|73.61| 73.61| 84.21 |73.61| 84.21 |96.54
> >> |Std.F.T|38.89| 0| 37.84 |5.55| 5.26 |98.77
> >>
> >> |Locality Rate (LR)|Edit 1|Edit 2|Edit 3|
> >> |--|----|---|--|
> >> |RefineViT(ours)|91.79|88.12|84.69|
> >> |WTE|90.48 |86.52 |85.176 |
> >> |Std.F.T|95.99| 83.84| 83.88
> >
> > - Result: RefineViT **consistently** maintains higher GR and LR than WTE and standard fine-tuning, indicating it preserves previous edits while maintaining locality, which demonstrates strong effectiveness in multiplex editing scenarios.

---

> > ### Comment · Reviewer_qQpH · 2025-08-04
> >
> > I thank the authors for conducting additional experiments. I think my concerns are partially addressed. But I am getting more confused by the authors' claim in the response to W1: they acknowledge the simplification and point to the limitation section (which is acceptable), they argue the method is "generally robust" because the edit only removes the direct contribution, leaving indirect effects intact, which I think is logically flawed.
> >
> > To my understanding, the core of the problem is in the identification stage, not the rectification stage. The utility function (which decides which heads are "problematic") is calculated based on a hypothetical representation derived from the linear model. If a head's primary negative impact is through its indirect influence on subsequent layers, the utility function will fail to detect it. The function is blind to the very effects which I am concerned with. Therefore, the claim that the approach is "robust to potential misidentification" is unsubstantiated. The method is only robust in the sense that it doesn't edit the indirect pathways, but it cannot be robust to misidentifying heads if its identification mechanism is blind to a major class of effects.

---

> ### Author Response · Authors · 2025-08-05
>
> We sincerely thank the reviewer for the thoughtful feedback.
>
> You are correct that the main limitation of our method lies in the identification stage: our utility function is based only on the first-order (direct) contributions of attention heads. As you pointed out, this means it may miss heads whose main effect is indirect, acting through later layers.
>
> We apologize for the confusion caused by the ambiguity in our previous response regarding this point. Our previous response focuses on the rectification stage. The rectification stage is minimally invasive, modifying only the direct contributions of selected heads and leaving indirect effects untouched. This design ensures that the edits do not introduce additional errors, but, as you noted, does not address errors from missed indirect contributors.
>
> We believe the first-order identification is still useful in practice for two reasons:
> - "Self-repair" in transformers makes indirect effects relatively weaker: Prior work [A, B] has shown that when a layer is ablated, later layers often compensate, making higher-order effects relatively weaker in many cases.
> - Indirect effects that propagate through later layers to the output. They may still be picked up by the direct effect of these later layers. In other words, any significant indirect effect from an earlier layer could manifest as a direct effect in one or more downstream heads. Thus, our utility function can capture and address these effects by targeting the direct contributions of downstream heads.
>
> We fully acknowledge that there can be cases where important indirect effects are not captured by our method, and these remain a limitation of our method and will be included in the Limitations section.
>
> Thank you again for highlighting this important point and helping us clarify our claims.
>
> [A] McGrath, T., Rahtz, M., Kramar, J., Mikulik, V., & Legg, S. The hydra effect: Emergent self-repair in language model computations. 2023.
> [B] Yossi Gandelsman, Alexei A. Efros, Jacob Steinhardt. Interpreting the Second-Order Effects of Neurons in CLIP. ICLR 2025

---

> ### Author Response · Authors · 2025-08-06
> **Further support our previous response regarding the limitation of considering only first‑order effects**
>
> To further support our previous response regarding the limitation of considering only first‑order effects, we conducted an additional experiment comparing our method with a variant that also accounts for indirect (second‑ and higher‑order) effects of attention heads during the identification stage.
>
> - **Experimental Setups.**
> > Instead of decomposing the image representation as a linear sum of the direct contributions of attention heads for identification and ablation, the variant measures each attention head’s influence by directly ablating its output in the forward pass, thereby capturing both direct and indirect effects. We evaluated both methods on the Waterbirds dataset, where for each method we use a single sample per edit, repeat three times to reduce randomness, and report the average accuracy and worst‑case accuracy when ablating the top‑T identified attention heads.
>
>
> - **Results:**
>   - The results (in tables below) show that the variant ("With indirect effects") performs marginally better for $T=1$. However, as $T$ increases, our method ("Direct effect only") consistently outperforms the variant in both average and worst-case accuracy. This suggests that while indirect effects may matter when ablating a single head, considering only direct effects yields better performance when more heads are removed.
>   - We believe this is because indirect effects are complex and intertwined, making it harder to achieve effective ablation when considering all orders. In contrast, focusing on direct effects allows more targeted and decoupled interventions.
>   - Additionally, our method is much more efficient: it requires only one forward pass, which can be reused for both training and testing, whereas the variant needs repeated forward passes for each head evaluation.
>   - Therefore, omitting indirect effects is both practical and effective. It achieves strong error correction performance while keeping computational cost low.
>
> > | Avg.Acc(%)            | Original | T=1        | T=3        | T=6        | T=9        | T=12       | T=15       |
> > | --------------------- | -------- | ---------- | ---------- | ---------- | ---------- | ---------- | ---------- |
> > | Direct effects only    | 72.85    | 72.75±0.77 | 75.33±1.88 | 78.21±1.21 | 79.37±1.93 | 80.03±1.22 | 79.92±0.74 |
> > | With indirect effects | 72.85    | 72.78±0.06 | 73.05±0.80 | 74.53±0.87 | 75.88±1.43 | 76.08±1.10 | 74.73±1.08 |
>
> > | Wst.Acc(%)            | Original | T=1        | T=3        | T=6        | T=9        | T=12       | T=15       |
> > | --------------------- | -------- | ---------- | ---------- | ---------- | ---------- | ---------- | ---------- |
> > | Direct effects only    | 45.62    | 44.16±2.04 | 50.98±5.31 | 60.92±2.18 | 64.10±5.02 | 67.63±4.65 | 66.98±6.79 |
> > | With indirect effects | 45.62    | 47.55±3.50 | 47.15±3.62 | 52.9±5.62  | 55.72±3.64 | 56.77±0.55 | 55.26±3.56 |
>
> Thank you again for bringing this important point to our attention. We hope our additional experiments and analysis have addressed your concern. If you have any further questions or would like more clarification, we are happy to provide it.

---

> > ### Comment · Reviewer_qQpH · 2025-08-07
> >
> > Thanks for the further information. My concern on this is addressed  and I will increase the score accordingly.

---

> > > ### Author Response · Authors · 2025-08-07
> > >
> > > We sincerely thank you for your thoughtful feedback and for raising the score. We greatly appreciate your recognition of our work and your constructive comments, which have helped us further clarify and strengthen the paper.

---

### Official Review · Reviewer_TBhL · 2025-07-03

**Clarity:** 3
**Significance:** 2
**Originality:** 2
**Rating:** 4
**Confidence:** 4

**Summary:**

This paper investigates the problem of model editing for Vision Transformers (ViTs), a largely underexplored topic compared to model editing in language models (LMs). While prior work on LMs often targets modifications in multi-layer perceptron (MLP) modules, the authors observe that ViT predictions are more influenced by multi-head self-attention (MSA) than by MLPs. Motivated by this insight, they propose RefineViT, a two-stage framework for editing ViTs. In Stage 1, RefineViT identifies spurious features by locating attention heads most responsible for misclassifications. In Stage 2, it corrects predictions by either ablating these attention heads (in simple cases) or learning a trainable projection matrix to adjust the image representation while preserving prediction locality. The method is validated across a broad range of benchmarks and ViT variants, including CLIP-ViT, demonstrating state-of-the-art performance in both edit success and edit locality.

**Questions:**

1. Can the head attribution process be made more robust to ambiguous or adversarial examples? The current process assumes that the utility scores effectively highlight spurious heads. Are there adversarial cases where this could be misleading?

2. How would the method generalize to multi-modal architectures like Flamingo or BLIP-2? Since RefineViT relies on decomposing image representations, could it be adapted to edit vision-language embeddings in multi-modal transformers?

3. How is performance affected by changing the number of attention heads ablated (T)? An ablation study on the sensitivity of the method to T (beyond the top-k selection) would be helpful, especially in low-resource or overfitting-prone regimes.

4. What are the limitations of applying the projection matrix universally across inputs? Since P(θ) is learned from a small set of error instances, could it harm performance on rare or edge cases not covered by the original edit scope?

5. Can RefineViT serve as a debugging tool to discover spurious concepts? It seems that the utility functions reveal which features lead to error. Could this be repurposed for interpretability or dataset bias detection?

6. Can the authors also report results on dense prediction tasks? Would the method generalize? E.g., the authors can study object detection or segmentation as a downstream task.

**Ethical Concerns:**

["NO or VERY MINOR ethics concerns only"]

**Final Justification:**

Based on the questions raised by the reviewers and the feedback from authors I am raising my score to 4.

**Limitations:**

Yes.

**Quality:**

3

**Strengths And Weaknesses:**

Strengths

1. The work is technically solid, with clear empirical grounding, rigorous derivations (e.g., utility functions for attention head attribution), and extensive experimental validation.

2. The paper is well-organized and well-written. The motivation is compelling, the diagrams (e.g., Fig. 1) are intuitive, and the theoretical analyses are easy to follow.

3.  Tackling model editing in ViTs is novel and timely, especially as ViTs become prevalent in real-world systems where fast and local error correction is valuable. The method has wide implications for robust deployment and interpretability of vision models.

4. The insight that editing MSA is more effective than MLPs in ViTs is novel. The formulation of utility-based attention head attribution and subsequent representation rectification via a projection matrix is both elegant and original.

Weaknesses

1. Reliance on utility-based heuristics: The head attribution stage relies on heuristics and utility scores that may be brittle in edge cases or less interpretable across tasks. A discussion on potential failure cases of this heuristic selection would be beneficial.

2. Assumes access to small labeled samples for editing: Although framed as data-efficient, the method still requires at least one labeled misclassified instance per error type, which may be non-trivial in open-set or online deployment settings.

3. Evaluation scope: The experiments are comprehensive, but limited to classification. It remains unclear how well this method generalizes to dense prediction tasks such as object detection or segmentation.

---

> ### Author Rebuttal · Authors · 2025-07-31
>
> We sincerely thank the reviewer for the positive feedback, particularly recognizing the technical soundness, clear presentation, and the novelty and timeliness of our ViT model editing approach. Below, we address all questions and weaknesses, supplemented with new supporting experiments.
>
> **Q1 & W1:** Given its reliance on utility-based heuristics. **(1)** Are there adversarial cases where this could be misleading? **(2)** Can the head attribution process be made more robust? **(3)** How interpretable?
>
> > **A:**  Point‑by‑point response:
> > **(1) Potential misleading cases.** Our method is inherently data-driven and thus sensitive to the quality of the editing sample, especially in the one-sample setting. Ambiguous or mislabeled samples can reduce the reliability of head attribution and the resulting edits.
> >
> >  To illustrate this, we conducted a new Waterbirds experiment comparing edits using a correct (“real”) misclassified sample versus an incorrect (“fake”) one. Editing with a real sample improves both average and worst-group accuracy, whereas a fake sample significantly degrades performance:
> >
> > |                   | Avg. Acc (%) | Wst. Acc (%)  |
> > | ----------------- | ------------ | ------------- |
> > | Original CLIP-ViT | 92.06        | 45.71         |
> > | One 'Real' sample     | 92.8 ± 1.18  | 53.9 ± 3.10   |
> > | One 'Fake' sample     | 78.53 ± 3.52 | 14.04 ± 12.36 |
> >
> > This demonstrates the **expected limitations under weak supervision** while confirming that the method behaves sensibly—**improving with informative signals and degrading with misleading ones.**
>
> > **(2) Robustness.** In scenarios where multiple samples are available, we mitigate brittleness **by leveraging four complementary utility scores** (Utility A–D, Appendix A, p.13) capturing different aspects of attention behavior. In the Waterbirds experiment (Appendix C, p.16), Grad-CAM visualizations and TextSpan-generated descriptions validate that the selected heads align with the intended utility functions.
>
> > **(3) Interpretability.** On Waterbirds, which contains strong spurious correlations (water/land backgrounds for bird-species classification), our utility scores consistently identify heads focusing on these background cues, as confirmed by TextSpan descriptions (Tables 4 & 5) and Grad-CAM visualizations (Fig. 7). These analyses demonstrate that our utility-based heuristics are both effective and interpretable.
>
>
>
> **W2.** The method requires at least one labeled misclassified instance per error type, which may be non-trivial in open-set or online deployment settings.
>
> >**A:** We acknowledge the need for at least one labeled misclassified instance to guide editing.
> >- However, **this is a minimal data requirement in the field of model editing.** As far as we know, there are no zero-shot model editing methods. Our method achieves effective editing using just a single such instance.
> >- In practice, model errors are typically discovered through observed prediction failures, making such instances naturally available. Hence, this requirement is reasonable and **aligns with real‑world deployment scenarios.**
>
> **W3. & Q6.** Can the method generalize to tense prediction tasks such as object detection or segmentation?
>
> > **A:** Thank you for the suggestion. **Yes, it can**.To assess generalization beyond classification, we follow the setup in [4] and apply our method to a **segmentation task** using a single edit sample.
> >- As shown in the table below, our method improves pixel-wise accuracy on similar images and even slightly boosts overall performance on the full dataset:
> > >
> > >| Avg. Pixel-Wise Accuracy | Similar Data | Overall |
> > >| ------------------------ | ------------ | ------- |
> > >| Original CLIP-ViT        | 72.29%       | 76.78%  |
> > >| CLIP-ViT-Edited (Ours)   | 74.50%       | 77.01%  |
> >
> > - To our knowledge, there are currently no established model editing methods for segmentation. This result serves as an initial demonstration of our method in this setting. In future work, we plan to build a benchmark for model editing in segmentation and explore more effective methods tailored to dense prediction tasks.
>
> **Q2.** Can the proposed method be generalized to multi-modal architectures like Flamingo or BLIP-2?
>
> >**A:** Our method focuses on editing visual transformers with standalone visual encoders. For multi-modal models like CLIP-ViT, which include an independent visual encoder, our method can be directly applied to the visual component.
>
> >In contrast, models such as BLIP-2 and Flamingo deeply fuse vision and language features via cross-attention, and their visual modules are typically frozen during training. Editing these tightly integrated architectures aligns more closely with large language model (LLM) editing than with visual transformer editing.
>
> >Therefore, adapting our method to these models is **non-trivial** and represents an interesting direction for future research.
>
> **Q3.** How is performance affected by changing the number of attention heads ablated?
>
>
> >**A:** We evaluated the effect of ablating different numbers of attention heads using the ViT Natural Image benchmark (class 954_582). As shown below, GR (target-class performance) **improves** as more heads are ablated, **peaks** around 6 heads, and then **slightly declines**, showing that ablating the right heads matters more than simply ablating more heads:
> >
> >   | N_ablated | 3   | 6   | 9   | 12  |
> >   | --- | --- | --- | --- | --- |
> >   | GR(%) | 73.78±13.08 | 86.67±2.30 | 84.89±2.03 | 84.44±2.78 |
> >
> > Importantly, the number of heads ablated is **not a fixed hyperparameter** in our method.
> >
> >   - In single-sample settings, $T$ refers to the number of heads actually ablated. We select heads whose utility scores exceed a fixed threshold.
> >   - In multi-sample settings (Appendix A, Page 13), **$T$ denotes the size of the candidate pool**, i.e., the number of top-ranked heads selected per utility score. The final ablation set is determined adaptively by applying a validation utility (as described in Appendix, Page 14, “Summary of the Procedure”), so the number ablated is typically smaller than $T$.
> >
> >Furthermore, the **sensitivity analysis** in Figure 8 (Appendix) shows that performance stabilizes when $T ≥ 10$ in multi-sample settings, indicating that beyond this candidate pool size, increasing $T$ has minimal effect as the most impactful heads are already included.
>
>
> **Q4.** The limitations of applying the projection matrix universally? Could it harm performance?
>
>
> >**A:** Applying a projection matrix universally may risk unintended changes, but our method is **specifically designed to minimize such effects while maintaining strong edit performance.**
>
> >- In model editing, three primary goals are typically considered: (1) edit success, (2) generalization, and (3) **locality**—the latter reflecting how well the edit preserves performance on unrelated data. In practice, **edits often involve trade-offs**, and to our knowledge, no method guarantees zero degradation on out-of-scope data.
> >- To address this, we introduce a **locality loss** to reduce such side effects explicitly. We report Locality Rate (LR) in Figures 4 and 5, showing that our method achieves a more favorable balance between edit success and preservation of unrelated behavior than prior approaches.
> >- Additionally, to assess the impact of using a projection matrix as the update mechanism, we compare it to LoRA, a commonly used alternative. As shown in Figure 14 (Appendix, Page 22), our projection-based strategy preserves both effectiveness and locality better than the LoRA-version baseline.
>
>
> **Q5.** Can RefineViT serve as a debugging tool to discover spurious concepts?
>
> >**A:** Yes, it can. As shown in Appendix C (Page 16), we evaluate our method on the Waterbirds dataset, which contains strong spurious correlations between bird type and background. Tables 4 and 5 list the attention heads identified by our utility scores, along with descriptions from TextSpan. Grad-CAM visualizations in Figure 7 further confirm that these heads attend to background regions, indicating our method’s ability to identify spurious features driving incorrect predictions.

---

> > ### Comment · Reviewer_TBhL · 2025-08-08
> >
> > thank you for your replies and your feedback to the other reviewers concerns.  I have no further questions.  Based on the discussion I am raising my score to 4.

---

> > > ### Author Response · Authors · 2025-08-08
> > >
> > > We greatly appreciate your detailed review, constructive feedback, and your decision to raise the score after reading our rebuttal. Thank you for the time and attention you’ve dedicated to our work.

---

> ### Author Response · Authors · 2025-08-05
>
> Thank you for reading our rebuttal. Did our response sufficiently address your concerns? We’d be happy to clarify or discuss any remaining questions you might have.

---

> ### Author Response · Authors · 2025-08-07
>
> We would like to clarify a minor issue in our previous response to Q1 & W1. The accuracy values we initially reported were calculated using both the target classes from Waterbirds and unrelated ImageNet data. The correct approach is to compute accuracy using only the target classes.
>
> Below, we provide the updated results. These results are consistent with our original claims and more clearly demonstrate our main point: mislabeled or ambiguous samples in the single-sample editing setting can reduce the reliability of attention head identification and the effectiveness of the resulting edits.
>
> |                   | Avg. Acc (%) | Wst. Acc (%)  |
> |-------------------|--------------|---------------|
> | Original CLIP-ViT | 72.85        | 45.62         |
> | One 'Real' sample | 79.37 ± 1.14 | 64.10 ± 5.02  |
> | One 'Fake' sample | 58.85 ± 4.88 | 14.04 ± 12.36 |
>
> As a gentle reminder, we would also be grateful to know whether our responses have sufficiently addressed your concerns. Please let us know if there are any remaining concerns or if you would like further clarification.

---

### Decision · Program_Chairs · 2025-09-17

**Decision:**

Accept (poster)

**Comment:**

The paper proposes a method for model editing for vision transformers. The authors address classification tasks, and identify the self-attention heads that lead the most to incorrect predictions. The authors then propose to counter the effect of these heads by applying a linear transformation to the embeddings that minimally affected unrelated samples, whilst also approximating the output obtained when these localised heads are removed.

Reviewers appreciated the insight from the authors that editing self-attention layers is more effective for vision transformers, as opposed to MLPs in transformers for language. Moreover, reviewers felt that this is a relevant, well-motivated problem and the paper is clear and well-written.

The main concerns of the reviewers were well-addresed during the rebuttal. Please update the final camera-ready with this.